# Unbiasing fermionic quantum Monte Carlo with a quantum computer

William J. Huggins[1✉], Bryan A. O'Gorman[2], Nicholas C. Rubin[1], David R. Reichman[3], Ryan Babbush[1] & Joonho Lee[1,3✉]

Interacting many-electron problems pose some of the greatest computational challenges in science, with essential applications across many fields. The solutions to these problems will offer accurate predictions of chemical reactivity and kinetics, and other properties of quantum systems[1–4]. Fermionic quantum Monte Carlo (QMC) methods[5,6], which use a statistical sampling of the ground state, are among the most powerful approaches to these problems. Controlling the fermionic sign problem with constraints ensures the efficiency of QMC at the expense of potentially significant biases owing to the limited flexibility of classical computation. Here we propose an approach that combines constrained QMC with quantum computation to reduce such biases. We implement our scheme experimentally using up to 16 qubits to unbias constrained QMC calculations performed on chemical systems with as many as 120 orbitals. These experiments represent the largest chemistry simulations performed with the help of quantum computers, while achieving accuracy that is competitive with state-of-the-art classical methods without burdensome error mitigation. Compared with the popular variational quantum eigensolver[7,8], our hybrid quantum-classical computational model offers an alternative path towards achieving a practical quantum advantage for the electronic structure problem without demanding exceedingly accurate preparation and measurement of the ground-state wavefunction.

The complexity of finding an accurate solution of the Schrödinger equation seemingly grows exponentially with the number of electrons in the system. This fact has greatly hindered progress towards an efficient means of accurately calculating ground-state quantum mechanical properties of complex systems. Over the last century, substantial research effort has been devoted to the development of new algorithms for solution of this many-electron problem. At present, all available general-purpose methods can be grouped into two categories: (1) methods that scale exponentially with system size while yielding numerically exact answers, and (2) methods for which the cost scales polynomially with system size, but that are only approximate by construction. Approaches in this second category are the only methods that can feasibly be applied to large systems at present. The accuracy of the solutions obtained by these methods is often unsatisfactory and is almost always difficult to assess.

Quantum computing has arisen as an alternative model for the calculation of quantum properties that may complement, and potentially surpass, classical methods in terms of efficiency[9,10]. Although the ultimate ambition of this field is to construct a universal fault-tolerant quantum computer[11], the experimental devices of today are limited to noisy intermediate-scale quantum (NISQ) computers[12]. NISQ algorithms for the computation of ground states have largely centred around the variational quantum eigensolver (VQE) framework[7,8], which necessitates coping with optimization difficulties, measurement overhead and circuit noise. As an alternative, algorithms based on imaginary-time evolution have been put forward, which, in principle, avoid the optimization problem[13,14]. However, because of the non-unitary nature of imaginary-time evolution, one must resort to heuristics to achieve reasonable scaling with system size. New strategies that avoid these limitations may help to enable the first practical quantum advantage in fermionic simulations. In this work, we propose and experimentally demonstrate a class of quantum-classical hybrid algorithms that offer a different route to addressing these challenges. We do not attempt to represent the ground-state wavefunction using our quantum processor, choosing instead to use it to guide a quantum Monte Carlo (QMC) calculation performed on a classical coprocessor. Using this approach, our experimental demonstration surpasses the scale of previous experimental work on quantum simulation in chemistry[15–17].

## Theory and algorithms

QMC approaches[5,6] target the exact ground-state wavefunction, $|\Psi_0\rangle$, of a many-body Hamiltonian, $\hat{H}$, via imaginary-time evolution of an initial state $|\Phi_0\rangle$ with a non-zero overlap with $|\Psi_0\rangle$:

$$|\Psi_0\rangle \propto \lim_{\tau \to \infty} |\Psi(\tau)\rangle, \quad |\Psi(\tau)\rangle \equiv e^{-\tau\hat{H}}|\Phi_0\rangle, \tag{1}$$

[1]Google Quantum AI, Mountain View, CA, USA. [2]Berkeley Quantum Information & Computation Center, University of California, Berkeley, CA, USA. [3]Department of Chemistry, Columbia University, New York, NY, USA. ✉e-mail: whuggins@google.com; jl5653@columbia.edu

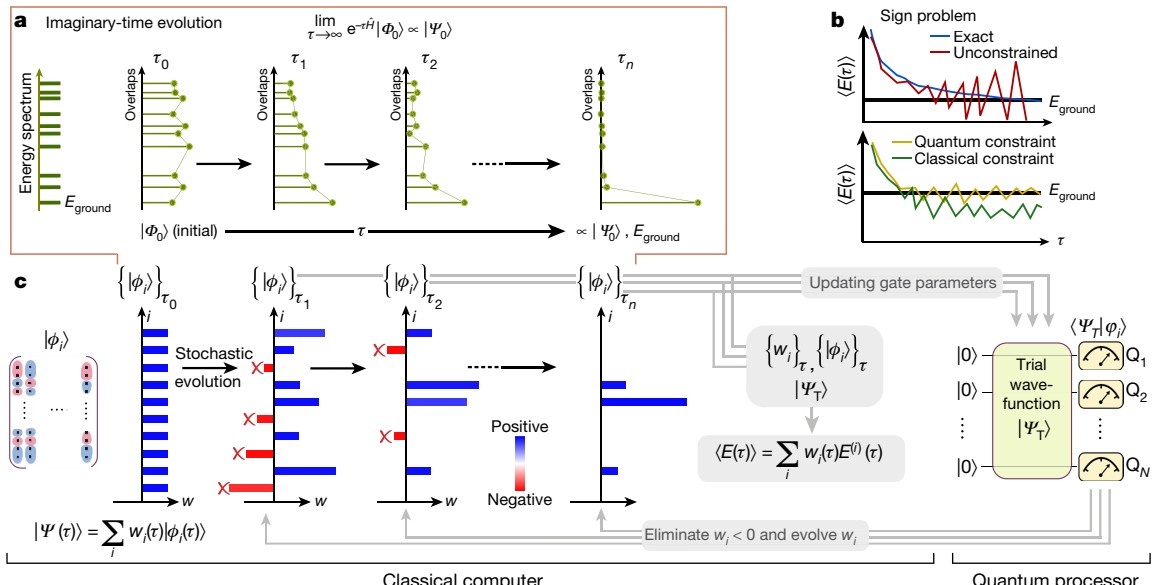

**Fig. 1 | Imaginary-time evolution, sign problem and our quantum-classical hybrid algorithm. a**, Depiction of the imaginary-time evolution, which shows an exponential convergence to the ground state as a function of imaginary time, $\tau$. **b**, Illustration of the fermionic sign problem. Exact deterministic imaginary-time evolution and an unconstrained QMC calculation, which is exact on average but has a signal-to-noise ratio that diverges with increasing $\tau$ due to the sign problem (top). Constrained QMC calculations with classical and quantum constraints. The use of quantum constraint helps to reduce the bias that is non-negligible when using the classical constraint (bottom). **c**, Overview of the QC-QMC

algorithm. Stochastic wavefunction samples, represented as $\{|\phi_i\rangle\}_\tau$, are evolved in time along with associated weights $\{w_i\}_\tau$. Throughout the time evolution, queries to the quantum processor about the overlap value between the quantum trial wavefunction $|\Psi_T\rangle$ and a stochastic wavefunction sample $\{|\phi_i\rangle\}_\tau$ are made while updating the gate parameters to describe $\{|\phi_i\rangle\}_\tau$. Our quantum processor uses $N$ qubits to efficiently estimate the overlap, which is then used to evolve $w_i$ and to discard stochastic wavefunction samples with $w_i < 0$. Finally, observables, such as $\langle E(\tau)\rangle$, are computed on the classical computer using overlap queries to the quantum processor (Supplementary Section C).

where $\tau$ is imaginary time and $|\Psi(\tau)\rangle$ denotes the time-evolved wavefunction from $|\Phi_0\rangle$ by $\tau$ (Fig. 1a). In QMC, the imaginary-time evolution in equation (1) is implemented stochastically, which can enable a polynomial scaling algorithm to sample an estimate for the exact ground-state energy by avoiding the explicit storage of high-dimensional objects, such as $\hat{H}$ and $|\Psi_0\rangle$. The ground-state energy, $E_{ground} = E(\tau = \infty)$, is estimated from averaging a time series of $\{E^{(i)}(\tau)\}$, given by a weighted average over $M$ statistical samples,

$$\langle E(\tau)\rangle = \sum_{i=1}^{M} w_i(\tau)E^{(i)}(\tau), \qquad (2)$$

where $E^{(i)}(\tau)$ is the $i$th statistical sample for the energy and $w_i(\tau)$ is the corresponding normalized weight for that sample at imaginary time $\tau$. Although formally exact, such a stochastic imaginary-time evolution algorithm will generically run into the fermionic sign problem[18], which manifests as a result of alternating signs in the weights of each statistical sample used in equation (2). In the worst case, the fermionic sign problem causes the estimator of the energy in equation (2) to have exponentially large variance (Fig. 1b, top), necessitating that one averages exponentially many samples so as to obtain a target precision. Accordingly, exact, unbiased QMC approaches are only applicable to small systems[19–21] or those lacking a sign problem[22].

The sign problem can be controlled to give an estimator of the ground-state energy with polynomially bound variance by imposing constraints on the imaginary-time evolution of each statistical sample represented by a trial wavefunction, $|\phi_i(\tau)\rangle$. These constraints (which include prominent examples such as the fixed node[6,23] and phaseless approximations[24,25]) are imposed by demanding that the overlaps of the trial wavefunction ($|\Psi_T\rangle$) (where T denotes trial) with the stochastic samples ($|\phi_i(\tau)\rangle$) remain positive during the imaginary-time propagation. Although constrained QMC calculations are typically much more accurate than those using the bare trial wavefunction directly (Fig. 1b,

bottom), the remaining bias of the constrained QMC results is wholly determined by the choice of the trial wavefunction. Imposing these constraints necessarily introduces a potentially significant bias in the final ground-state energy estimate, which can be removed in the limit that the trial wavefunction approaches the exact ground state. Alternatively, the bias can be removed by releasing the constraints during propagation, at the expense of suffering an uncontrolled sign problem[26].

Classically, computationally tractable options for trial wavefunctions are limited to states such as a single mean-field determinant (for example, a Hartree–Fock state), a linear combination of mean-field states, a simple form of the electron–electron pair (two-body) correlator (usually called a Jastrow factor) applied to mean-field states or some other physically motivated transformations applied to mean-field states, such as backflow approaches[27]. On the other hand, any wavefunction that can be prepared with a quantum circuit is a candidate for a trial wavefunction on a quantum computer, including more general two-body correlators. These trial wavefunctions will be referred to as 'quantum' trial wavefunctions.

At present, there is no efficient classical algorithm to estimate (to additive error) the overlap between $|\phi_i(\tau)\rangle$ and various quantum trial wavefunctions $|\Psi_T\rangle$, such as unitary coupled-cluster with singles and doubles[28], qubit coupled-cluster methods[29], wavefunctions constructed by adiabatic state preparation[30] or the multiscale entanglement renormalization ansatz[31]. This is true even when $|\phi_i(\tau)\rangle$ is simply a computational basis state or a Slater determinant. As quantum computers can efficiently approximate $\langle\Psi_T|\phi_i(\tau)\rangle$, there is a potential quantum advantage in this task, as well as its particular use in QMC. This offers a different route towards quantum advantage in ground-state fermion simulations compared with VQE, which instead seeks an advantage in the variational energy evaluation. We expand on this discussion of quantum advantage in Supplementary Section F.

Our quantum-classical hybrid QMC algorithm (QC-QMC) utilizes quantum trial wavefunctions while performing the majority of

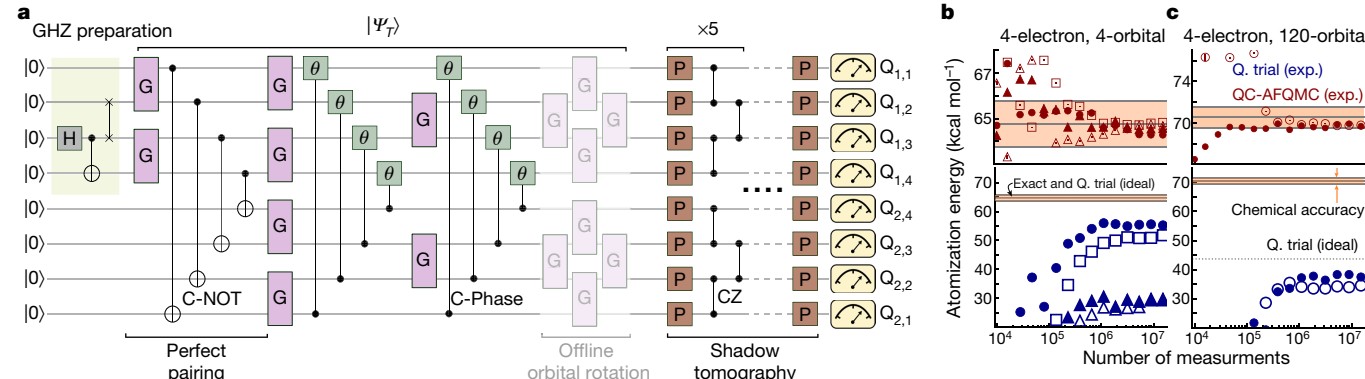

**Fig. 2 | 8-qubit experiment. a**, Circuit used for the 8-qubit $H_4$ experiment over a 2 × 4 qubit grid (from $Q_{1,1}$ to $Q_{2,1}$) on the Sycamore quantum processor[43]. In the circuit diagram, H denotes the Hadamard gate, G denotes a Givens rotation gate (generated by XX + YY), P denotes a single-qubit Clifford gate and $|\Psi_T\rangle$ denotes the quantum trial wavefunction. Note that the 'offline' orbital rotation is not present in the actual quantum circuit because it is handled via classical post-processing, as discussed in Supplementary Section C. **b, c**, Convergence of the atomization energy of $H_4$ as a function of the number of measurements. A minimal basis set (STO-3G) with four orbitals total from four independent experiments (exp.) (**b**) and a quadruple-zeta basis set (cc-pVQZ) with 120 orbitals total from two independent experiments (**c**). The different symbols in

**b** and **c** show independent experimental results. Note that the ideal (that is, noiseless) atomization energy of the quantum trial (Q. trial) in **b** is precisely on top of the exact one and that the QC-AFQMC energy would likewise be exact in the absence of noise. For the system in **c**, QC-AFQMC with this quantum trial would yield an error of 0.2 kcal mol$^{-1}$ despite a much larger error in the variational energy of the quantum trial. Further note that the quantum resource used in **c** is 8 qubit, but, as shown in Supplementary Section C, our algorithm enables the addition of 'virtual' electron correlation in a much larger Hilbert space. The top panels of **b** and **c** magnify the energy range near the exact answer. See Extended Data Tables 1–8 for the raw data for **b, c**, as well as other relevant data.

imaginary-time evolution on a classical computer, and is summarized in Fig. 1c. In essence, on a classical computer one performs imaginary-time evolution for each wavefunction statistical sample, $|\phi_i(\tau)\rangle$, and collects observables such as the ground-state energy estimate, $E^{(i)}(\tau)$. During this procedure, a constraint associated with the quantum trial wavefunction is imposed to control the sign problem. To perform the constrained time evolution, the only quantity that needs to be calculated on the quantum computer is the overlap between the trial wavefunction, $|\Psi_T\rangle$, and the statistical sample of the wavefunction at imaginary time $\tau$, $|\phi_i(\tau)\rangle$. Although our approach applies generally to any form of constrained QMC, here we discuss an experimental demonstration of the algorithm that uses an implementation of QMC known as auxiliary-field QMC (AFQMC), which will be referred to as QC-AFQMC (see Methods for more details). As a single determinant mean-field trial wavefunction is the most widely used classical form of the trial function for AFQMC owing to its efficiency, here we use 'AFQMC' to denote AFQMC with a mean-field trial wavefunction.

## Discussion

As the first example, in Fig. 2 we illustrate the quantum primitive used to perform the experiment on an $H_4$ molecule involving 8 qubits (see Methods for more details). Our eight spin-orbital quantum trial wavefunction consists of a valence bond wavefunction known as a perfect pairing state[32,33] and a hardware-efficient quantum circuit[15] with an offline single-particle rotation, which would be classically difficult to use as a trial wavefunction for AFQMC. The state preparation circuit in Fig. 2a shows how this trial wavefunction can be efficiently prepared on a quantum computer.

In this 8-qubit experiment, we consider $H_4$ in a square geometry with side lengths of 1.23 Å and its dissociation into four hydrogen atoms. This system is often used as a test bed for electron correlation methods in quantum chemistry[34,35]. We perform our calculations using two Gaussian basis sets: the minimal (STO-3G) basis set[36] and the correlation consistent quadruple-zeta (cc-pVQZ) basis set[37]. The latter basis set is of the size and accuracy required to make a direct comparison with laboratory experiments. When describing the ground state of this system, there are two equally important, degenerate mean-field states. This makes AFQMC with a single mean-field trial wavefunction highly

unreliable. In addition, a method often referred to as a 'gold standard' classical approach (that is, coupled-cluster with singles, doubles and perturbative triples, CCSD(T)[38]) also performs poorly for this system.

In Table 1, the difficulties of AFQMC and CCSD(T) are well illustrated by comparing their atomization energies with exact values in two different basis sets. Both approaches show errors that are significantly larger than 'chemical accuracy' (1 kcal mol$^{-1}$). The variational energy of the quantum trial reconstructed from experiment has a bias that can be as large as 33 kcal mol$^{-1}$. The noise on our quantum device makes the quality of our quantum trial far from that of the ideal (that is, noiseless) ansatz, as shown in Fig. 2b, c, resulting in an error as large as 10 kcal mol$^{-1}$ in the atomization energy. Nonetheless, QC-AFQMC reduces this error significantly, and achieves chemical accuracy in both bases. Notably, we achieve this accuracy even in the larger basis, where the variational energy of the quantum trial in the absence of noise is far from exact.

As shown in Supplementary Section C, for the larger basis set we obtain a residual 'virtual' correlation energy by using the quantum resources on a smaller number of orbitals to unbias an AFQMC calculation on a larger number of orbitals, with no additional overhead to the quantum computer. This capability makes our implementation competitive with state-of-the-art classical approaches. Similar virtual correlation energy strategies have been previously discussed within the framework of VQE[39], but, unlike our approach, those strategies come with a significant measurement overhead. To unravel the QC-AFQMC results on $H_4$ further, in Fig. 2b, c we illustrate the evolution of trial

## Table 1 | Atomization energy (kcal mol$^{-1}$) of $H_4$

|  | Exact | AFQMC | CCSD(T) | Q. trial | QC-AFQMC |
|---|---|---|---|---|---|
| 4-orbital | 64.7 | 62.9 | 59.6 | 55.2 | 64.3 |
| 120-orbital | 70.5 | 68.6 | 71.9 | 37.4 | 69.7 |

Data for quantum trial (Q. trial; experiment), AFQMC (classical), QC-AFQMC (experiment), CCSD(T) (classical 'gold standard') and exact results for minimal (STO-3G; 4-orbital) and quadruple-zeta (cc-pVQZ; 120-orbital) bases. Both of these last two experiments use 8 qubits. The statistical error of AFQMC and QC-AFQMC is less than 0.05 kcal mol$^{-1}$ and therefore is not shown. Note that, as shown in Supplementary Section E, Q. trial results vary significantly run-to-run, while QC-AFQMC results are nearly identical run-to-run (showcasing the noise resilience of QC-AFQMC).

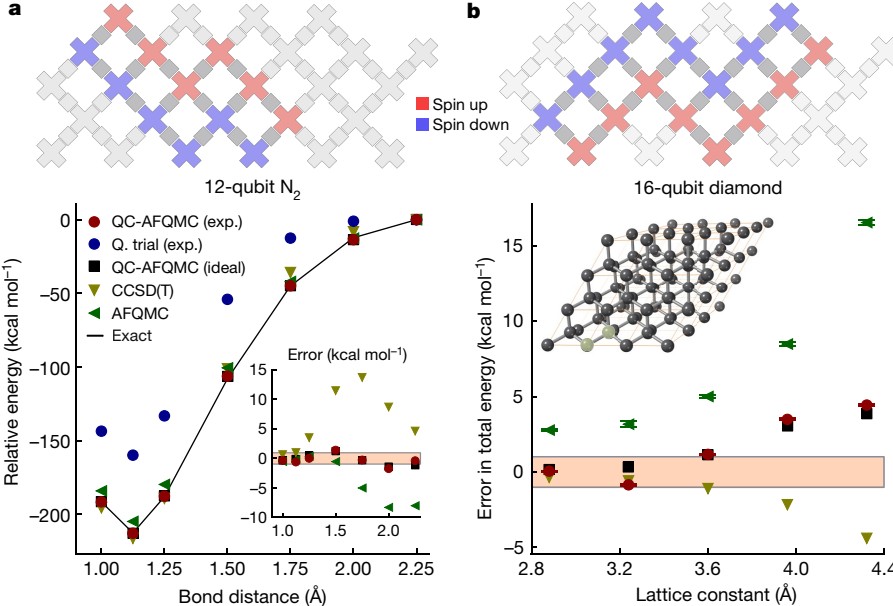

**Fig. 3 | 12-qubit and 16-qubit experiments. a**, Circuit layout showing spin-up and spin-down qubits for the 12-qubit experiment (top). Potential energy surface of $N_2$ in a triple-zeta basis set (cc-pVTZ[37]; 60 orbital) (bottom). The relative energies are shifted to zero at 2.25 Å. Inset shows the error in total energy relative to the exact results in kcal mol$^{-1}$. The shaded region in the inset shows the bounds for chemical accuracy (1 kcal mol$^{-1}$). Neither the variational energy of the quantum trial nor the statistical error bars of the AFQMC

methods are visible on this scale. **b**, Circuit layout showing spin-up and spin-down qubits for the 16-qubit experiment (top). Error in total energy as a function of lattice constant of diamond in a double-zeta basis (DZVP-GTH; 26 orbitals) (bottom). The shaded region shows the bounds for chemical accuracy. Our quantum trial results are not visible on this scale. Inset shows a supercell structure of diamond in which two highlighted atoms form the minimal unit cell. See Extended Data Tables 9, 10 for the raw data for **a**, **b**, respectively.

and QC-AFQMC energies as a function of the number of measurements made on the device. Despite the presence of significant noise in approximately $10^5$ measurements, QC-AFQMC achieves chemical accuracy while coping with a sizeable residual bias in the underlying quantum trial.

Next, we move to a larger example, $N_2$, which requires a total of 12 qubits in our quantum experiment. Here a simpler quantum trial is used for QC-AFQMC by taking just the valence bond part of the wavefunction depicted in Fig. 2a. We examine the potential energy surface of $N_2$ from compressed to elongated geometries, which is another common benchmark problem for classical quantum chemistry methods[35,40]. In Fig. 3a, the QC-AFQMC result is shown for the calculations performed in a triple-zeta basis (cc-pVTZ) set[37], which corresponds to a 60-orbital or 120-qubit Hilbert space. All examined methods, CCSD(T), AFQMC and QC-AFQMC, perform well near the equilibrium geometry, but CCSD(T) and AFQMC deviate from the exact results significantly as the bond distance is stretched. As a result, the error for 'gold standard' CCSD(T) can be as large as 14 kcal mol$^{-1}$, and the error for AFQMC with a classical trial wavefunction can be as large as $-8$ kcal mol$^{-1}$. The error in the QC-AFQMC computation ranges from $-2$ kcal mol$^{-1}$ to 1 kcal mol$^{-1}$ depending on the bond distance. Thus, although we do not achieve chemical accuracy with QC-AFQMC, we note that, even with a simple quantum trial wavefunction, we produce energies that are competitive with state-of-the-art classical approaches. Idealized (that is, noiseless) VQE experiments for the same trial wavefunction would yield similar results to our quantum trial results Fig. 3a (within 4.5 kcal mol$^{-1}$), which are much worse than our QC-AFQMC results with an error as large as 50 kcal mol$^{-1}$.

Finally, we present a 16-qubit experiment result for the ground-state simulation of a minimal unit cell (two-atom) model of periodic solid diamond in a double-zeta basis set (DZVP-GTH[41]; 26 orbitals). Although at this level of theory the model exhibits significant finite-size effects and does not predict the correct experimental lattice constant, we aim to illustrate the utility of our algorithm in materials science applications.

We emphasize that this is the largest quantum simulation of chemistry on a quantum processor so far (detailed resource counts and comparison with prior works are available in Extended Data Tables 11, 12). We again use the simple perfect pairing state as our quantum trial wavefunction and demonstrate the improvement over a range of lattice parameters compared with classical AFQMC and CCSD(T) in Fig. 3b. There is a substantial improvement in the error going from AFQMC to QC-AFQMC, showing the increased accuracy due to better trial wavefunctions. At the same time, QC-AFQMC performed using the idealized quantum trial produces results comparable to our experimental energies, suggesting that the error in our QC-AFQMC energies is mainly due to the use of an insufficiently accurate trial wavefunction rather than experimental error. Our accuracy is limited by the simple form of our quantum trial and yet we achieve accuracy nearly on a par with the classical gold standard method, CCSD(T).

## Conclusion and outlook

In summary, we propose a scalable, noise-resilient quantum-classical hybrid algorithm that seamlessly embeds a special-purpose quantum primitive into an accurate quantum computational many-body method, namely QMC. Our work offers a computational strategy that effectively unbiases fermionic QMC approaches by leveraging state-of-the-art quantum information tools. We have realized this algorithm for a specific QMC algorithm known as AFQMC, and demonstrated its performance in experiments as large as 16 qubit on a NISQ processor, producing electronic energies that are competitive with state-of-the-art classical quantum chemistry methods. Our algorithm also enables the incorporation of the electron correlation energy outside the space that is handled by the quantum computer without increasing quantum resources or measurement overheads. In Supplementary Section F, we discuss issues related to asymptotic scaling and the potential for quantum advantage in our algorithm. Although we have yet to achieve practical quantum advantage over available classical algorithms, the

flexibility and scalability of our proposed approach in the construction of quantum trial functions, and its inherent noise resilience, promise a path forward for the simulation of chemistry in the NISQ era and beyond.

*Note added in proof:* After this work was nearly complete, a theory paper by Yang et al. appeared on arXiv[42], describing a quantum algorithm for assisting real-time dynamics with unconstrained QMC.

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

# Methods

## Wavefunction overlap estimation

In this work, we estimate the overlap between the trial wavefunction and the statistical samples using a technique known as shadow tomography[43,44]. Experimentally, this entails performing randomly chosen measurements of a reference state related to $|\Psi_T\rangle$ before beginning the QMC calculation, yielding the representation of $|\Psi_T\rangle$ in the computational basis for subsequent overlap evaluations. In this formulation of QC-QMC, there is no need for the QMC calculation to iteratively query the quantum processor, despite the fact that the details of the statistical samples are not determined in advance. By disentangling the interaction between the quantum and classical computer we avoid feedback latency, an appealing feature on early NISQ platforms that comes at the cost of requiring potentially expensive classical post-processing (see Supplementary Section D for more details). Furthermore, our algorithm naturally achieves some degree of noise robustness, as explained in Supplementary Section D, because the quantity directly used in QC-QMC is the ratio between overlap values, which is inherently resilient to the estimates of the overlaps being rescaled. We highlight the challenges posed by the need to measure wavefunction overlaps precisely and the trade-offs involved in the use of shadow tomography (see also Supplementary Section D), while giving our perspective on the most promising paths forward.

## Phaseless constraints in AFQMC

In AFQMC, the $|\phi_i(\tau)\rangle$ take the form of Slater determinants in arbitrary single-particle bases, enabling us to express the energy estimator (presented in Supplementary equation (3)) in terms of a modest number of wavefunction overlaps that we can evaluate efficiently on the quantum processor (Supplementary Section C). The phaseless constraint is imposed to control the sign problem and, likewise, only requires calculating the overlaps between $|\Psi_T\rangle$ and $|\phi_i(\tau)\rangle$, as detailed in Supplementary equation (6). AFQMC has been shown to be accurate in a number of cases even with classically available trial wavefunctions[45,46]; however, the bias incurred from the phaseless constraint cannot be overlooked.

## Quantum processor

The experiments in this work were carried out on the Google 54-qubit quantum processor known as Sycamore[47]. The circuits were compiled using hardware-native conditional Z gates with typical error rates of $\approx 0.5\%$ (ref. [48]).

## Data availability

The datasets generated and/or analysed during the current study are available from the corresponding authors on reasonable request. Source data are provided with this paper.

## Code availability

We used available packages such as Q-Chem[49] and Cirq (see https://github.com/quantumlib/Cirq for details on obtaining the source code); more details are available in Supplementary Section E. Other codes used herein are available from the corresponding authors on reasonable request.

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

**Acknowledgements** We thank members of the Google Quantum AI theory team and F. Malone for helpful discussions. J.L. and D.R.R. acknowledge the support of NSF CHE-1954791. B.O. is supported by a NASA Space Technology Research Fellowship and the NSF QLCI program through grant number OMA-2016245. The quantum hardware used for this experiment was developed by the Google Quantum AI hardware team, under the direction of A. Megrant, J. Kelly and Y. Chen. Theoretical foundations for device calibrations were provided by the physics team lead by V. Smelyanskiy. Initial data collection was enabled by cloud access to these devices as part of Google Quantum AI's Quantum Computing Service Early Access Program. P. Roushan and C. Neill from the Google team helped to execute the experiment on hardware and design figures.

**Author contributions** J.L. conceived the quantum-classical hybrid QMC algorithm, performed QMC calculations and, with contribution from others, drafted the manuscript. W.J.H. proposed the use of shadow tomography and designed the experiment, with contributions from others. B.O. helped with theoretical analysis and the compilation of circuits. N.C.R. helped with the presentation of figures. J.L. and R.B. managed the scientific collaboration. All authors participated in discussions, writing the manuscript and analysis of the data. J.L. and W.J.H. contributed equally to this work.

**Competing interest** The authors declare no competing interests.

**Additional information**
**Correspondence and requests for materials** should be addressed to William J. Huggins or Joonho Lee.

**Extended Data Table 1 | Experimental data of variational energy for $H_4$ in STO-3G with partitioned tomography**

| $N_{\text{Cliffords}}$ | repeat 1 | repeat 2 | repeat 3 | repeat 4 |
|---|---|---|---|---|
| 10 | -1.800644 | -1.764747 | -1.813274 | -1.658202 |
| 16 | -1.823041 | -1.802192 | -1.840494 | -1.730591 |
| 28 | -1.906644 | -1.839835 | -1.843326 | -1.746749 |
| 47 | -1.925654 | -1.888527 | -1.860863 | -1.809656 |
| 80 | -1.909567 | -1.869456 | -1.887139 | -1.846339 |
| 136 | -1.930880 | -1.902309 | -1.889992 | -1.879164 |
| 229 | -1.944249 | -1.921523 | -1.903710 | -1.890947 |
| 387 | -1.947362 | -1.934682 | -1.910477 | -1.901883 |
| 652 | -1.952416 | -1.939853 | -1.912790 | -1.905250 |
| 1100 | -1.955544 | -1.944651 | -1.915073 | -1.909122 |
| 1856 | -1.955028 | -1.945966 | -1.909558 | -1.908038 |
| 3129 | -1.953877 | -1.947763 | -1.913386 | -1.908835 |
| 5276 | -1.954697 | -1.947323 | -1.912284 | -1.909315 |
| 8896 | -1.954930 | -1.947458 | -1.913889 | -1.913068 |
| 15000 | -1.954356 | -1.948894 | -1.913894 | -1.913082 |

Variational energy of $|\Psi_T\rangle$ from four independent repeated partitioned shadow tomography experiments with a different set of random Cliffords for $H_4$, STO-3G (minimal basis). If the experiment was perfect (i.e., no circuit noise), then the variational energy should approach –1.969512.

**Extended Data Table 2 | Experimental data of variational energy for $H_4$ in STO-3G with unpartitioned shadow tomography**

| $N_{\text{Cliffords}}$ | repeat 1 | repeat 2 | repeat 3 | repeat 4 |
|---|---|---|---|---|
| 10 | -1.643633 | -1.798261 | -1.671065 | -1.462214 |
| 16 | -1.720721 | -1.848279 | -1.747911 | -1.645383 |
| 28 | -1.816519 | -1.911599 | -1.786704 | -1.737425 |
| 47 | -1.867034 | -1.920776 | -1.777655 | -1.819957 |
| 80 | -1.887030 | -1.901445 | -1.825170 | -1.844560 |
| 136 | -1.924619 | -1.930137 | -1.845217 | -1.858595 |
| 229 | -1.929421 | -1.933710 | -1.847781 | -1.871717 |
| 387 | -1.940266 | -1.936080 | -1.851352 | -1.880681 |
| 652 | -1.936394 | -1.937956 | -1.860513 | -1.878550 |
| 1100 | -1.935905 | -1.936406 | -1.875337 | -1.881012 |
| 1856 | -1.938452 | -1.938114 | -1.877807 | -1.884442 |
| 3129 | -1.939407 | -1.939186 | -1.880363 | -1.887409 |
| 5276 | -1.936669 | -1.939222 | -1.882466 | -1.890464 |
| 8896 | -1.937593 | -1.938921 | -1.872013 | -1.888485 |
| 15000 | -1.938364 | -1.939795 | -1.871097 | -1.887922 |

Same as Extended Data Table 1 but for the unpartitioned shadow tomography experiments.

**Extended Data Table 3 | Experimental data of variational energy for $H_4$ in cc-pVQZ with partitioned tomography**

| $N_{\text{Cliffords}}$ | repeat 1 | repeat 2 |
|---|---|---|
| 10 | -1.996118 | -1.658351 |
| 16 | -1.988746 | -1.557607 |
| 28 | -2.009853 | -1.873220 |
| 47 | -2.019875 | -1.976545 |
| 80 | -2.026756 | -1.983726 |
| 136 | -2.034241 | -2.005448 |
| 229 | -2.030444 | -2.045285 |
| 387 | -2.051324 | -2.052698 |
| 652 | -2.053210 | -2.056238 |
| 1100 | -2.059021 | -2.054032 |
| 1856 | -2.059920 | -2.053114 |
| 3129 | -2.057736 | -2.053142 |
| 5276 | -2.060762 | -2.054276 |
| 8896 | -2.060786 | -2.053847 |
| 15000 | -2.059437 | -2.054775 |

Variational energy of $|\Psi_T\rangle$ from four independent repeated partitioned shadow tomography experiments with a different set of random Cliffords for $H_4$, cc-pVQZ (a quadruple-zeta basis). If the experiment was perfect (i.e., no circuit noise), then the variational energy should approach −2.069364.

**Extended Data Table 4 | Experimental data of variational energy for $H_4$ in STO-3G with unpartitioned tomography**

| $N_{\text{Cliffords}}$ | repeat 1 | repeat 2 |
|---|---|---|
| 10 | -1.794532 | -1.961018 |
| 16 | -1.864535 | -1.963510 |
| 28 | -1.971853 | -2.015256 |
| 47 | -2.028933 | -2.025942 |
| 80 | -2.022666 | -2.029521 |
| 136 | -2.044745 | -2.032204 |
| 229 | -2.050697 | -2.036077 |
| 387 | -2.055859 | -2.038768 |
| 652 | -2.054068 | -2.042764 |
| 1100 | -2.055576 | -2.047633 |
| 1856 | -2.054740 | -2.049588 |
| 3129 | -2.055636 | -2.051308 |
| 5276 | -2.056442 | -2.052641 |
| 8896 | -2.056741 | -2.052579 |
| 15000 | -2.056641 | -2.051843 |

Same as Extended Data Table 3 but for the unpartitioned shadow tomography experiments.

**Extended Data Table 5 | Experimental data of QC-AFQMC energy for $H_4$ in STO-3G with partitioned shadow tomography**

| $N_{\text{Cliffords}}$ | repeat 1 | repeat 2 | repeat 3 | repeat 4 |
|---|---|---|---|---|
| 10 | -1.96943(5) | -1.98295(6) | -1.96873(6) | -1.9724(1) |
| 16 | -1.97376(5) | -1.97385(6) | -1.97175(4) | -1.9672(1) |
| 28 | -1.97019(3) | -1.97083(4) | -1.97267(4) | -1.97343(8) |
| 47 | -1.97033(2) | -1.96931(3) | -1.97261(4) | -1.97400(7) |
| 80 | -1.97016(3) | -1.97398(4) | -1.97061(4) | -1.97038(6) |
| 136 | -1.97042(2) | -1.97240(4) | -1.97054(4) | -1.96821(5) |
| 229 | -1.97046(2) | -1.97090(2) | -1.96931(4) | -1.96844(5) |
| 387 | -1.97019(2) | -1.97076(2) | -1.97010(4) | -1.96831(5) |
| 652 | -1.97030(2) | -1.97013(2) | -1.96929(4) | -1.96861(4) |
| 1100 | -1.96928(2) | -1.96958(2) | -1.96931(4) | -1.96882(5) |
| 1856 | -1.96942(2) | -1.96964(1) | -1.96974(4) | -1.96909(5) |
| 3129 | -1.96914(2) | -1.96948(2) | -1.96933(4) | -1.96922(4) |
| 5276 | -1.96879(2) | -1.96947(2) | -1.96914(4) | -1.96944(5) |
| 8896 | -1.96877(2) | -1.96959(2) | -1.96918(4) | -1.96952(4) |
| 15000 | -1.96877(2) | -1.96964(2) | -1.96922(4) | -1.96941(4) |

AFQMC energy using $|\Psi_T\rangle$ from four independent repeated partitioned shadow tomography experiments with a different set of random Cliffords for $H_4$, STO-3G (minimal basis). The exact ground-state energy is −1.969512. The numbers in parentheses indicate the statistical error of the AFQMC energy.

**Extended Data Table 6 | Experimental data of QC-AFQMC energy for $H_4$ in STO-3G with unpartitioned shadow tomography**

| $N_{\text{Cliffords}}$ | repeat 1 | repeat 2 | repeat 3 | repeat 4 |
|---|---|---|---|---|
| 10 | -2.0058(1) | -1.97058(9) | -1.9712(1) | -1.9823(2) |
| 16 | -1.9907(1) | -1.96982(8) | -1.97094(9) | -1.9869(1) |
| 28 | -1.98318(7) | -1.96711(4) | -1.97036(9) | -1.97288(6) |
| 47 | -1.97642(5) | -1.96859(3) | -1.9823(1) | -1.97291(6) |
| 80 | -1.97430(4) | -1.97010(5) | -1.9833(1) | -1.96990(5) |
| 136 | -1.97131(3) | -1.96846(3) | -1.97343(8) | -1.97025(6) |
| 229 | -1.97114(2) | -1.96934(3) | -1.97253(8) | -1.96970(6) |
| 387 | -1.96995(2) | -1.97006(3) | -1.97059(8) | -1.96981(6) |
| 652 | -1.96982(3) | -1.96995(3) | -1.97024(7) | -1.96980(7) |
| 1100 | -1.96975(3) | -1.97054(3) | -1.96955(7) | -1.96958(7) |
| 1856 | -1.96940(3) | -1.97017(3) | -1.96886(7) | -1.96975(7) |
| 3129 | -1.96926(3) | -1.97013(3) | -1.96884(7) | -1.96984(7) |
| 5276 | -1.96940(3) | -1.96999(3) | -1.96931(7) | -1.96968(7) |
| 8896 | -1.96950(3) | -1.97011(3) | -1.96918(8) | -1.96954(7) |
| 15000 | -1.96952(3) | -1.97022(3) | -1.96943(7) | -1.96930(7) |

Same as Extended Data Table 5 but for the unpartitioned shadow tomography experiments.

**Extended Data Table 7 | Experimental data of QC-AFQMC energy for H$_4$ in cc-pVQZ with partitioned shadow tomography**

| $N_{\text{Cliffords}}$ | repeat 1 | repeat 2 |
|---|---|---|
| 10 | -2.10573(9) | -2.1461(3) |
| 16 | -2.10766(9) | -2.1214(5) |
| 28 | -2.1095(1) | -2.1344(3) |
| 47 | -2.1107(2) | -2.1214(1) |
| 80 | -2.11063(5) | -2.1313(2) |
| 136 | -2.11039(6) | -2.1220(1) |
| 229 | -2.11044(6) | -2.11312(5) |
| 387 | -2.11120(7) | -2.11141(4) |
| 652 | -2.11026(7) | -2.11176(7) |
| 1100 | -2.11090(4) | -2.11105(4) |
| 1856 | -2.11067(3) | -2.11131(4) |
| 3129 | -2.11055(6) | -2.11120(5) |
| 5276 | -2.11105(4) | -2.11090(4) |
| 8896 | -2.11119(5) | -2.11092(6) |
| 15000 | -2.11081(3) | -2.11098(4) |

AFQMC energy using $|\Psi_T\rangle$ from four independent repeated partitioned shadow tomography experiments with a different set of random Cliffords for H$_4$, cc-pVQZ (a quadruple-zeta basis). The exact ground-state energy is –2.11216599. The numbers in parentheses indicate the statistical error of the AFQMC energy.

**Extended Data Table 8 | Experimental data of QC-AFQMC energy for H$_4$ in cc-pVQZ with unpartitioned shadow tomography**

| $N_{\text{Cliffords}}$ | repeat 1 | repeat 2 |
|---|---|---|
| 10 | -2.1188(2) | -2.1070(1) |
| 16 | -2.1146(1) | -2.1080(1) |
| 28 | -2.10942(9) | -2.11169(9) |
| 47 | -2.10951(6) | -2.11108(7) |
| 80 | -2.1111(1) | -2.11219(7) |
| 136 | -2.11100(4) | -2.11064(6) |
| 229 | -2.11105(4) | -2.11218(6) |
| 387 | -2.11069(3) | -2.11197(7) |
| 652 | -2.11068(4) | -2.11159(8) |
| 1100 | -2.11048(4) | -2.11180(5) |
| 1856 | -2.1109(1) | -2.11206(6) |
| 3129 | -2.11092(6) | -2.11198(5) |
| 5276 | -2.11015(3) | -2.11186(5) |
| 8896 | -2.11045(3) | -2.11220(5) |
| 15000 | -2.11040(4) | -2.11182(5) |

Same as Extended Data Table 7 but for the unpartitioned shadow tomography experiments.

**Extended Data Table 9 | Raw data for N$_2$ potential energy surface for seven bond distances (*R*)**

| R(Å) | Exact | CCSD(T) | Quantum trial (exp.) | Quantum trial (ideal) | AFQMC | QC-AFQMC (exp.) | QC-AFQMC (ideal) |
|---|---|---|---|---|---|---|---|
| 1.000 | -109.366398 | -109.365383 | -109.017231 | -109.025925 | -109.3672(3) | -109.36697(7) | -109.36685(9) |
| 1.125 | -109.399981 | -109.398412 | -109.043176 | -109.053858 | -109.4003(3) | -109.40094(7) | -109.4001(1) |
| 1.250 | -109.360887 | -109.355280 | -109.000672 | -109.009988 | -109.3603(4) | -109.36085(8) | -109.3604(2) |
| 1.500 | -109.233325 | -109.215012 | -108.874636 | -108.880510 | -109.2342(3) | -109.23109(9) | -109.2309(2) |
| 1.750 | -109.132826 | -109.110942 | -108.808418 | -108.810112 | -109.1408(2) | -109.13325(8) | -109.1332(1) |
| 2.000 | -109.080654 | -109.066772 | -108.790143 | -108.790084 | -109.0939(2) | -109.08341(7) | -109.08298(7) |
| 2.250 | -109.061147 | -109.053758 | -108.788486 | -108.792041 | -109.07392(8) | -109.06177(7) | -109.06293(6) |

Note that the energy of our quantum trial here is obtained from a single set of experiments which may vary significantly run-to-run. Quantum trial (ideal) indicates the variational energy of the trial wavefunction evaluated exactly assuming that there is no noise in the circuit execution. Similarly, QC-AFQMC (ideal) represents the QC-AFQMC data obtained with the ideal quantum trial wavefunction.

**Extended Data Table 10 | Raw data for the diamond cold curve for five lattice constants (*R*)**

| R(Å) | Exact | CCSD(T) | Quantum trial (exp.) | Quantum trial (ideal) | AFQMC | QC-AFQMC (exp.) | QC-AFQMC (ideal) |
|---|---|---|---|---|---|---|---|
| 2.880 | -9.545911 | -9.546464 | -9.121081 | -9.426260 | -9.5415(1) | -9.54582(5) | -9.54555(4) |
| 3.240 | -10.229155 | -10.230100 | -8.625292 | -10.092197 | -10.2241(3) | -10.23051(7) | -10.22837(5) |
| 3.600 | -10.560477 | -10.562229 | -10.277938 | -10.405326 | -10.5525(2) | -10.55861(8) | -10.55864(6) |
| 3.960 | -10.700421 | -10.703884 | -10.368882 | -10.528044 | -10.6869(2) | -10.6949(1) | -10.6956(1) |
| 4.320 | -10.744089 | -10.751103 | -10.222206 | -10.571136 | -10.7177(3) | -10.73701(9) | -10.73810 (9) |

Note that the energy of our quantum trial here is obtained from a single set of experiments which may vary significantly run-to-run. Note that these energies include the Madelung constant. Quantum trial (ideal) indicates the variational energy of the trial wavefunction evaluated exactly assuming that there is no noise in the circuit execution. Similarly, QC-AFQMC (ideal) represents the QC-AFQMC data obtained with the ideal quantum trial wavefunction.

**Extended Data Table 11 | Resource counts for the QC-AFQMC experiments realized in this work**

| Experiment | # Qubits | # CZ Gates (State Prep) | # CZ Gates (Total) | Circuit Depth |
|---|---|---|---|---|
| Hydrogen (Partitioned) | 8 | 36 | 66 | 52 |
| Hydrogen (Unpartitioned) | 8 | 36 | 99 | 67 |
| Nitrogen | 12 | 22 | 92 | 53 |
| Diamond | 16 | 34 | 160 | 65 |

**Extended Data Table 12 | Resource estimates from prior fermionic simulations using gate-model quantum computers on more than four qubits**

| Experiment | Reference | # Qubits | # 2q Gates |
|---|---|---|---|
| $BeH_2$ | [59] | 6 | 5 ($U_{ENT}$) |
| $H_2O$ | [60] | 5 | 6 ($XX(\theta)$) |
| Hydrogen | [61] | 12 | 72 ($\sqrt{i}\mathrm{SWAP}$) |
| Diazene | [61] | 10 | 50 ($\sqrt{i}\mathrm{SWAP}$) |
| Hubbard, interacting (8-site) | [62] | 16 | 608 ($\sqrt{i}\mathrm{SWAP}$) |
| Hubbard, non-interacting (8-site) | [62] | 16 | 1568 ($\sqrt{i}\mathrm{SWAP}$) |

For the two Hubbard model experiments we distinguish between dynamics simulated for an interacting versus a non-interacting model. $N = 8$ indicates an eight site linear lattice with open boundary conditions. $U_{ENT}$ is a nearest-neighbour cross-resonance style gate and $XX(\theta)$ is a $\exp(-i\theta\sigma_x^i\sigma_x^j/2)$. As far as we are aware, these are the largest simulations using a gate-model quantum computer targeting fermionic ground states or dynamics. For references, see Supplementary Information.