## [Peer Review File · Nature]

Manuscript Title: Unbiasing Fermionic Quantum Monte Carlo with a Quantum Computer

Reviewer Comments & Author Rebuttals

Reviewer Reports on the Initial Version:

Referee #1 (Remarks to the Author):

Overview

In this manuscript, the authors set forth a new hybrid quantum-classical approach that leverages a quantum trial wave function generated by a quantum computer for determining the ground state energies of multireference species such as H₄ and N₂ using Auxiliary Field Quantum Monte Carlo (AFQMC). Classical AFQMC is one of the most accurate classical electronic structure techniques for probing the states of matter and properties of molecules. However, like virtually all many-body electronic structure techniques, it scales exponentially - in its exact form - with system size due to the fermion sign problem. One way of curbing the fermion sign problem to obtain a polynomially-scaling algorithm is to use a phaseless constraint that prevents samples from possessing negative weights. This constraint necessitates the use of an accurate trial wave function; the more accurate this trial wave function is, the more accurate and less biased the final results are.

Taking a very novel angle, this work encodes more accurate trial wave functions that cannot easily be manipulated on classical computers on a quantum computer and uses them to constrain AFQMC random walks performed on a classical computer. The overlap of this trial wavefunction with different orthogonal determinants that can be used to compute the overall overlap and system energy is calculated through shadow tomography measurements and then employed within the classical algorithm. Because the quantum trial wavefunctions are more accurate, the final system energies obtained are now more accurate than what would be obtained (although not exact!) using standard classical wavefunctions, leading to a supposed quantum advantage.

That said, I find this paper to be extremely novel - it is the first I am aware of to propose how to do hybrid quantum-classical computing based on quantum Monte Carlo techniques. The idea to use quantum trial wavefunctions that are more accurate than classical ones for constraints is also genuinely new. Nonetheless, I question the practical quantum advantage of the algorithm. Namely, what is the true scaling of the shadow tomography measurements as a function of both system size and correlation? Is its exponential more friendly than the exponential that accompanies using multideterminant expansions classically? This issue needs to be addressed before one can truly assert quantum advantage (beyond the theoretical advantage of using more accurate trial wave functions).

Substantive Questions and Concerns

1. Key Question: Is this work trading one exponential scaling for another? What is the cost of accurately measuring the trial wavefunction to a given precision (preferably 10^{-12} Hartree)? Does this also inevitably scale exponentially? (Wouldn't it have to if the wave function is fully entangled?) Is the scaling of this measurement less severe than that of multideterminant expansions, which while not exact, are pretty accurate at the end of the day? For instance, you talk of taking tens of millions of trial wave function measurements; I wouldn't think you need tens of millions of determinants (definitely not nonorthogonal determinants) to solve H₄, particularly in a small basis set.
2. Doesn't feeding in tens of millions of overlaps into your classical algorithm for each step of the propagation and walker start to scale in the trillions of operations? Doesn't this count start to necessitate a very powerful classical computer as well?
3. The clarity of the article would be improved if you clearly stated what the constraints on the imaginary time evolution have to do with the overlap with the trial wave function on page 2. Even

though those familiar with QMC would know this, this isn't explained for those unfamiliar in an intuitive way and is instead left to the equations in the Appendix.

4. It would also be clearer if the authors included at least a 1-sentence clarification/intuitive explanation in the main text of how the overlaps are computed without needing to know the on-the-fly determinants that arise during the QMC propagation. The equations in the Appendices clarify, but the way this is worded in the main text can lead to doubts.

5. Since you must preprogram your quantum trial wavefunction, you could never prepare the exact trial wave function without knowing it ahead of time, meaning that your algorithm would never be exact. That said, what is the hierarchy of increasingly more accurate wave functions you could program? And, what would be the cost of preparing and measuring these wave functions to "see" this increased accuracy? How challenging would it be to program QMC-derived multideterminant expansions so that you could iterate upon your solutions to get around this exactness issue?

6. How large is your phase problem with the more accurate quantum trial wave function without employing the phaseless approximation? Much like in Sharma's recent progress work on free projection with coupled cluster wave functions, one could ask how accurate your answers are and how much projection you need without employing the constraint.

7. May you break down the sources of your systematic errors QC-AFQMC errors compared to the exact results? How much is this from an inaccurate trial wave function? How much is this from potential measurement inaccuracies (if significant)? Are there other sources?

8. What is the actual time and resource expenditure for performing these quantum measurements?

Minor Edits

1. Equation 1's equations would be clearer if they were combined.

2. In the abstract, this sentence could be made less winding: "...popular variational quantum eigensolver in terms of potential towards the first practical quantum advantage in ground state many-electron calculations."

3. This phrase is misworded: "...quantity that is directly is the ratio..."

4. Under Table 1, I believe you want to define QT as quantum trial in the caption, but didn't explicitly do so and vacillate between using quantum trial and QT in your Table and descriptions. Moreover, you could do a better job explicitly stating that the quantum trial is the wave function generated on your quantum computer. This was not clear to me for over a paragraph.

5. Given the discussion of coupled cluster expansions, you should cite Sandeep Sharma's recent papers on free projection AFQMC with a coupled cluster trial, and its improved accuracy.

Referee #2 (Remarks to the Author):

A new quantum monte carlo (QMC) approach for quantum computing in quantum chemistry has been developed, this methodology has not been used in near term quantum computing before, and what is even more impressive, has not been implemented on the actual hardware before. The methodology has a relevant place for benefiting from a quantum computer by introducing more accurate trial wavefunction ansatzes (e.g. based on unitary coupled cluster techniques) and evaluating classically challenging overlaps between walkers' wavefunctions and the trial wavefunction. This computational approach can be compared with previously developed the variational quantum eigensolver (VQE) where similar trial wavefunction ansatzes can be used. The advantage of the current approach is that it does not require a challenging optimization of trial wavefunction parameters and yet seems to provide more accurate results (although I have a question regarding this below).

The considered chemical systems are indeed challenging for state-of-the-art classical methods and the obtained results are indeed impressive and will be of interest for the scientific community.

Overall, the manuscript is written quite clearly and can be understood by a general reader, but there are few places where more precise statements are needed (more detailed below).

I think the paper has a potential to be published in Nature after addressing a few questions below:

1) Trial wavefunction within result on Fig. 3a, was it obtained with or without optimization of parameters (amplitudes for generators) in the VQE fashion? I suspect it is without optimization, then its results are not a good measure for advantage of the QMC approach over VQE. It would be great to see the comparison between QMC and VQE using the same ansatz for the trial function for some system even using a classical simulator. What is not clear for me is whether QMC surpasses VQE in accuracy for the same ansatz, but to answer this, one needs to optimize the ansatz parameters in the VQE case.

2) As was pointed out by the authors they didn't reach the chemical accuracy for N₂ in all geometry configurations, which is a challenging system, especially considering that the one-electron basis was cc-pVTZ, but what I am wondering is what would be needed to overcome this problem at least in theory? What I feel is missing in the manuscript (& supplementary info) is a discussion on what are the conditions for obtaining the exact full-CI energy via QMC approach? Some qualitative statements are made that walkers wavefunctions and the trial wavefunction affect results, but it is not clear to me what limits the accuracy in the current approach? Technically, if we have any trial wavefunction that is not orthogonal to the exact wavefunction we should get the right answer, but in reality there are limitations on the walkers to be single determinant, which limits the overall accuracy, I assume. Some clear discussions of these questions are needed, they should address the question what one should do to systematically converge to the exact answer. Like in VQE one can make exponentially large ansatz to converge to the exact answer.

3) Weights update: this part is not clear, how the weights are selected and why needs to be explained more (at least in an appendix). For example, around eq B5, I would suggest to state clearly what is the equation for the weight at $\tau + d\tau$ as a function of previous time weights, and explain why such an update is used.

4) What is the role of the Hubbard-Stratonovich transformation in imaginary time propagation? Why is it needed on top of the trotterization?

5) In selection of the trial wavefunction ansatz, the authors do not mention any non-UCC-derived approaches, even though it was shown that qubit-based techniques can be more efficient (fewer 2-qubit gates and lower circuit depth) than those obtained from the fermionic algebras. I think this needs to be mentioned.

6) Choosing the shadow tomography over introducing the transformations into the eigen-frame of P_{\pm} operators has not been justified. It would be interesting to know why the authors believe that random circuits are more efficient than those that bring P_{\pm} operators into the Z-basis (computational basis).

Referee #3 (Remarks to the Author):

Review

In "Unbiasing Fermionic Quantum Monte Carlo with a Quantum Computer", by Huggins et al. they present a new way of utilize NISQ era devices. The hybrid-quantum classical approach leverages the intellectual technology of Monte Carlo and the hardware technology of Google Quantum AI

Labs. From the Monte Carlo vantage, the paper relies on auxiliary field quantum Monte Carlo (AFQMC) and aim to alleviate the errors made by the phase less approximation. The walkers are updated following a phaseless approximation whereby negative weight walkers can be removed. The error due to the phaseless approximation depends on the trial wave function. This leads to the quantum computational part of the algorithm. The quantum algorithm for this methods requires measuring overlaps between the walkers and the quantum computer's trial wave function. This is a novel concept that I was excited to learn about and is precisely the type of manuscript that should be published in Nature.

The measurement AFQMC scheme does not require any feedback loops between the quantum device and the classical hardware. This scheme still relies on preparing a good quantum state (which is QMA-complete). They utilize a specific classical ansatz (perfect pairing) and, in some cases, perform further variational optimization of the state. The obstacle of state preparation still looms over this approach. Despite the advances provide, this is not a silver bullet for demonstrating quantum supremacy via simulation methods. (The authors thoughtfully address this point the conclusions and in Appendix F.) Also of interest is the use of the relatively new technique of shadow tomography. This allows the authors to reach state of the art quantum simulations of molecular systems larger than all previous quantum-classical hybrid schemes.

I found the paper well written and well thought through. The paper is a good representation of the modern efforts in quantum computing and this paper fits into the Nature magazine goals quite nicely. I saw no typos and the paper, its logic, and results all seems correct.

One question I have: other Monte Carlo methods (VMC, DMC, and FCIQMC) aren't commented on extensively. I am not as familiar with AFQMC as the other methods. The other approaches don't require extracting overlaps. It isn't clear from reading the main text and perusing the many appendices if there is direct application to these other methods QMC methods. For instance, the trial wave function determines the nodes of DMC but its not clear that extracting the overlaps will help improve the nodal structure.

One mild suggestion I have: in the main text, the description of the technique is quite terse. It is unclear from reading the main text how the hybrid algorithm work (even at a cursory level). Equations B3 and B6 seem to underlie the main idea of the whole paper. I think including them e.g. in the caption for Figure 2 or before turning to "Results and discussion" would be ideal from a reader's perspective.

Overall recommendation: publish with optional edits.

Referee #4 (Remarks to the Author):

The manuscript by Huggins et al. provides a new take on the use of quantum computer prototypes as co-processors in quantum chemistry calculations. The authors show that the use of a quantum device as part of a quantum Monte Carlo calculation performed on a classical computer can, when combined with a few additional (classical) tricks, provide an advantage over the common classical approach. The results are contrasted to a number of other common approaches in quantum chemistry, both classical (CCSD(T)) and quantum (VQE) and surpass these.

The paper is written rather technical and comprehensive, visible not only in the large number of references, but also the very extensive appendix, which is frequently referenced. This style of presentation, while technically very complete and accurate, makes it harder to follow the presentation and somewhat obscures the achievement of this new algorithmic method.

The frequent reference to the present work being the largest quantum chemistry simulation ever performed on quantum hardware does not signal a major breakthrough, however, but rather an

incremental improvement. In fact, as shown and openly stated in the paper, the bare results and noise levels from the machine prevent it from producing reliable results - it is simply the structure of the (mostly classical) algorithm calling the quantum device, that makes the results useable in the examples demonstrated. As Figure 3.b shows, larger systems (despite only including 2 atoms and 26 orbitals - much less than the 60 orbitals in 3.a) no longer shown chemical accuracy.

In summary, while the specific algorithmic workflow using the quantum chip is novel, there is nothing surprising or a breakthrough in these results. The heavy lifting remains on the classical computing side and the quantum chip's noisy data is evaluated in a way that makes the calculation less susceptible to noise. Lastly, given the small number of gates (Fig 2.a) and qubits (8-16) used in these experiments, I do wonder, why the authors do not simulate their quantum computer to make a statement about suitability and scalability under reduced noise levels and improved hardware.

Ultimately, I cannot recommend publication in Nature and would recommend to submit to a more specialized quantum journal such as npj quantum information.

Author Rebuttals to Initial Comments:

Referee #1:

Overview

In this manuscript, the authors set forth a new hybrid quantum-classical approach that leverages a quantum trial wave function generated by a quantum computer for determining the ground state energies of multireference species such as H_4 and N_2 using Auxiliary Field Quantum Monte Carlo (AFQMC). Classical AFQMC is one of the most accurate classical electronic structure techniques for probing the states of matter and properties of molecules. However, like virtually all many-body electronic structure techniques, it scales exponentially - in its exact form - with system size due to the fermion sign problem. One way of curbing the fermion sign problem to obtain a polynomially-scaling algorithm is to use a phaseless constraint that prevents samples from possessing negative weights. This constraint necessitates the use of an accurate trial wave function; the more accurate this trial wave function is, the more accurate and less biased the final results are.

Taking a very novel angle, this work encodes more accurate trial wave functions that cannot easily be manipulated on classical computers on a quantum computer and uses them to constrain AFQMC random walks performed on a classical computer. The overlap of this trial wavefunction with different orthogonal determinants that can be used to compute the overall overlap and system energy is calculated through shadow tomography measurements and then employed within the classical algorithm. Because the quantum trial wavefunctions are more accurate, the final system energies obtained are now more accurate than what would be obtained (although not exact!) using standard classical wavefunctions, leading to a supposed quantum advantage.

We thank the referee for the succinct overview of our work. The essence of the AFQMC implementation of our algorithm is well described in the above paragraphs. We also emphasize that our work is generally applicable to other projector QMC approaches in addition to AFQMC and that our focus on AFQMC is just one concrete instantiation of the broader idea.

That said, I find this paper to be extremely novel - it is the first I am aware of to propose how to do hybrid quantum-classical computing based on quantum Monte Carlo techniques. The idea to use quantum trial wavefunctions that are more accurate than classical ones for constraints is also genuinely new.

We are happy that the referee found our work to be highly novel.

Nonetheless, I question the practical quantum advantage of the algorithm. Namely, what is the true scaling of the shadow tomography measurements as a function of both system size and correlation?

We thank the referee for this question. This is an important technical aspect in practical applications of our approach that could be overlooked in the main text. We added stronger pointer sentences in the main text to draw interested readers' attention into pertinent sections in the Supplementary Information: *"In SI F, we discuss issues related to asymptotic scaling and the*

potential for quantum advantage in our algorithm. We highlight the challenges posed by the need to measure wavefunction overlaps precisely and the trade-offs involved in the use of shadow tomography (see also SI D3), while giving our perspective on the most promising paths forward.” Here we also provide a comprehensive answer to the reviewer’s question summarizing important points made in the Supplementary Information.

The classical postprocessing of shadow tomography is currently inefficient when used with AFQMC. We note possible routes to overcome this in Supplementary Information F such as leveraging different variants of shadow tomography based on different ensembles of random unitaries, similar to what is done for fermionic local observables.⁶⁸ Moreover, we want to emphasize that this classical postprocessing *is efficient* for other QMC flavors such as lattice Green’s function MC (GFMC) as noted in Supplementary Information F. We would like to highlight that the shadow tomography technique is not necessary for the workflow of QC-QMC. As described in Supplementary Information D, we employ shadow tomography instead of the more standard Hadamard test to remove the relatively high latency interaction between quantum and classical computers during QMC runs. Removing the online interaction was helpful for realizing our QC-QMC algorithm on an early NISQ machine. However, shadow tomography is not required in the long run and can be replaced by the Hadamard test. The Hadamard test approach has larger constant factors but better asymptotic scaling.

In both shadow tomography and Hadamard test formulations of our approach, understanding the scaling requires rather delicate considerations. Many of these considerations are not fully worked out, which speaks for our work’s novelty. We cover these topics in Supplementary Information D and F where such issues are discussed with more subtlety than space in the main text allows.

Setting the classical post-processing cost aside, shadow tomography can estimate the overlap between a walker (a statistical sample) and a trial state up to an additive error E with the measurement complexity that goes like $O(1/E^2)$ (neglecting logarithmic factors). The same measurement complexity can be obtained by the Hadamard test regardless of the particular flavor of QMC, but without any computationally intensive postprocessing (even for AFQMC). But again, using the Hadamard test comes at the cost of higher latency communication between CPU and QPU, which only introduces constant factors to the overall complexity but still proves onerous for currently available NISQ machines.

As described in Supplementary Information F, the overall measurement complexity can be connected to system size and correlation length. Because the overlap between a walker and a trial decays exponentially with system size, our overall asymptotic measurement complexity scales exponentially in system size to maintain a fixed relative error in the overlap (as opposed to additive error) **in the worst case**. Depending on the system size, this measurement complexity does not necessarily dominate the cost of QC-QMC, so in practice, one might not experience significant overhead due to this (e.g., we do not encounter a problem due to this for the instances simulated in our paper). We note that this exponential scaling is closely related to the exponential scaling of the overlap of any easy-to-prepare initial states with the ground state (a cost that is often overlooked when quantifying the cost of using the quantum phase estimation to prepare ground states). In fact, it is necessary that for some systems our method has exponential scaling since chemistry is QMA-Hard in the worst case.

All approaches, including VQE and phase estimation, have a similar exponential cost that

can manifest in the worst case. However, like with VQE and phase estimation, we do not expect QC-QMC to be bottlenecked by an exponential measurement complexity for all systems of interest. For example, if the correlation length is finite, then an efficient evaluation of the overlap is possible using the virtual correlation technique (see Supplementary Information C3) developed in this work. We discuss several other strategies to reduce this measurement complexity in Supplementary Information F and give more details about one particular route, later in this response document.

Is its exponential more friendly than the exponential that accompanies using multideterminant expansions classically? This issue needs to be addressed before one can truly assert quantum advantage (beyond the theoretical advantage of using more accurate trial wave functions).

Yes, our QC-QMC approach provides clear routes toward higher efficiency than what is usually observed using multideterminant expansions classically. The reason is that the measurement complexity due to the vanishing overlap can be ameliorated if complicated wavefunction forms (such as unitary coupled-cluster) are used for walkers. This implementation does not introduce any additional exponential-scaling overheads to the QC-QMC algorithm while significantly reducing the measurement complexity. We have added a sentence in the second paragraph of Supplementary Information F that emphasizes this point: *“This allows us to efficiently work with both trial and walker wavefunctions that would require an exponentially large multideterminant expansion to accurately represent with a classical computer.”*

We note that all known classical implementations of complicated walker wavefunctions currently introduce additional exponential-scaling overheads. As long as the measurement complexity does not dominate the cost, our overall algorithm will be observed as a polynomial-scaling algorithm that has great potential to go significantly beyond classical multideterminant expansions. We have added an additional sentence to the last paragraph of the main text that will help encourage the interested reader to read the technical discussion in Supplementary Information F.

Substantive Questions and Concerns

1. Key Question: Is this work trading one exponential scaling for another? What is the cost of accurately measuring the trial wavefunction to a given precision (preferably 10^{-6} Hartree)? Does this also inevitably scale exponentially? (Wouldn't it have to if the wave function is fully entangled?) Is the scaling of this measurement less severe than that of multideterminant expansions, which while not exact, are pretty accurate at the end of the day? For instance, you talk of taking tens of millions of trial wave function measurements; I wouldn't think you need tens of millions of determinants (definitely not nonorthogonal determinants) to solve H_4 , particularly in a small basis set.

Using the alternative Hadamard test formulation of QC-QMC allows us to measure the overlaps using a number of samples that depend (polynomially) only on the desired (additive) error, not the structure of the walker or the trial wavefunction. Nonetheless, an exponential-scaling measurement overhead caused by the required precision in the overlap

measurements may still bottleneck our QC-QMC in the worst-case scenarios. We would expect an exponential overhead for any method, in such cases, including all existing quantum algorithms because of the QMA-Hardness of the problem in general. However, we have explicated several ways to reduce such an overhead significantly so that for practical examples (e.g., those explored in our paper), such overhead may not be seen. By doing so, our QC-QMC approach offers the potential to use complicated quantum trial wavefunctions well beyond the limit of classical multideterminant expansions.

2. Doesn't feeding in tens of millions of overlaps into your classical algorithm for each step of the propagation and walker start to scale in the trillions of operations? Doesn't this count start to necessitate a very powerful classical computer as well?

We agree with the referee's assessment that the cost of such a large number of operations is considerable and that it could necessitate the use of a powerful classical computer as well. In particular, using an approach based on shadow tomography requires substantial classical post-processing. The classical postprocessing for each wavefunction overlap is inefficient when we choose the walkers to be arbitrary Slater determinants and use a form of shadow tomography based on random Clifford circuits. This steep requirement for classical resources is specific to the shadow tomography implementation of QC-AFQMC. The use of shadow tomography is not required in QC-AFQMC (or QC-QMC more broadly), so this is not necessarily the limitation of our algorithm.

Another factor that affects the total number of operations is that we need to evaluate these overlaps for each component of the classical shadow every time step for each walker. We take some comfort in the fact that the total number of walkers and the total projection time required does not scale with the size of the system. More importantly, though, this is a cost common to all projector Monte Carlo approaches including the classical algorithms.

QC-QMC can be implemented by using the Hadamard test to evaluate the overlap for each walker once at every timestep. In this formulation, the overlap is obtained by performing a simple average of some number of measurement results, a number which depends only on the target (additive) error in the overlap evaluation. The need to perform the classical postprocessing per-walker and per-timestep remains, but this step of the calculations is extremely simple and efficient. We have added a sentence emphasizing this point to the first paragraph of the subsection of Supplementary Information F that begins with the phrase "*Quantum advantage in the overlap estimation,*" reading "*In fact, the only post-processing required in this case is to compute a simple average over +1 and -1-valued measurement outcomes, regardless of the complexity of the states involved.*" This formulation minimizes the computational overhead pointed out by the referee. Given the practical tradeoffs between the classical and quantum computational costs, we expect that future works should be devoted to studying the various design choices available in QC-QMC to understand optimal protocols.

3. The clarity of the article would be improved if you clearly stated what the constraints on the imaginary time evolution have to do with the overlap with the trial wave function on page 2. Even though those familiar with QMC would know this, this isn't explained for those unfamiliar in an intuitive way and is instead left to the equations in the Appendix.

We thank the referee for this point. We modified the fourth paragraph of the section beginning “*Theory and algorithms*” (found on the left-hand side of page 2) to explicitly state that the overlap between a trial wavefunction and a walker wavefunction is used to constrain the imaginary time evolution by forcing the overlap to remain positive. The pertinent sentences now read “*These constraints (which include prominent examples such as the fixed node^{17,23} and phaseless approximations^{24,25}) are imposed by demanding that the overlaps of the trial wavefunction ($|\Psi_T\rangle$) with the stochastic samples ($|\phi_i(\tau)\rangle$) remain positive during the imaginary-time propagation.*”

4. It would also be clearer if the authors included at least a 1-sentence clarification/intuitive explanation in the main text of how the overlaps are computed without needing to know the on-the-fly determinants that arise during the QMC propagation. The equations in the Appendices clarify, but the way this is worded in the main text can lead to doubts.

We thank the referee for this point. Our manuscript now has the following sentence that clarifies this point in the second full paragraph on the left-hand side of page 3: “*Experimentally, this entails performing randomly chosen measurements of a reference state related to $|\Psi_T\rangle$ prior to beginning the QMC calculation, yielding the representation of $|\Psi_T\rangle$ in the computational basis for subsequent overlap evaluations.*”

5. Since you must preprogram your quantum trial wavefunction, you could never prepare the exact trial wave function without knowing it ahead of time, meaning that your algorithm would never be exact.

This is a good point. The accuracy of constrained QMC calculations is ultimately limited by the quality of the trial wavefunction. We have modified the text on the left-hand side of page two to emphasize this by adding the sentence “*While constrained QMC calculations are typically much more accurate than those using the bare trial wavefunction directly (see Fig. 1(b) bottom), the remaining bias of the constrained QMC results are wholly determined by the choice of the trial wavefunction.*”. However, we do have the option of making our approach exact by performing a release constraint calculation [S. Sorella, Phys. Rev. B 84, 241110 (2011)]. We have added a sentence to the end of that same paragraph that points out this possibility to the reader: “*Alternatively, the bias can be removed by releasing the constraints during propagation at the expense of suffering an uncontrolled sign problem.*²⁶” Our inexactness comes with the efficiency that will allow us to study chemical systems beyond the reach of available brute-force classical approaches.

That said, what is the hierarchy of increasingly more accurate wave functions you could program? And, what would be the cost of preparing and measuring these wave functions to “see” this increased accuracy? How challenging would it be to program QMC-derived multideterminant expansions so that you could iterate upon your solutions to get around this exactness issue?

These are excellent forward-looking questions. We mention examples of more sophisticated trial wavefunctions that could be used to increase the accuracy of our approach in our main text. These include unitary coupled-cluster states, qubit coupled-cluster states, and the multiscale entanglement renormalization ansatz. Your comment also prompted us to add a mention of

wavefunctions constructed using adiabatic state preparation to the main text (to the first partial paragraph on the left-hand side of page 3) and to the Supplementary Information (in the third-to-last paragraph of Supplementary Information F).

The unitary coupled-cluster state and its simplified versions are good candidates for systematically improving the trial wavefunction by adding more excitation operators. Similarly, approaches based on adiabatic state preparation can also be systematically improved by traversing an adiabatic path more slowly. Since our approach is new, the cost and impact of refining these wavefunctions in the context of QC-QMC is still a research-level question that we hope to explore ourselves soon. We note that while the state preparation would necessarily become more costly (in terms of quantum circuit depth) when using more complicated trial wavefunctions, our discussion and analysis of the measurement protocols is basically independent of the choice of trial wavefunction.

6. How large is your phase problem with the more accurate quantum trial wave function without employing the phaseless approximation? Much like in Sharma's recent progress work on free projection with coupled cluster wave functions, one could ask how accurate your answers are and how much projection you need without employing the constraint.

Given reduced phaseless approximation errors for chemical systems considered here, we expect that the phase problem is not so severe starting from the quantum trial wavefunctions. Unfortunately, we do not have the technical tools to perform free-projection AFQMC at the moment. Such calculations are carried out within closed-end random walks whereas our AFQMC code currently supports only open-end random walks. This is sufficiently different so that the work involved in extending our numerical results to free projection AFQMC is non-trivial. Prompted by this comment, we now mention the possibility of doing such numerics in the future in the second to last paragraph Supplementary Information F: "*Our prospects for observing a quantum advantage with these sophisticated wavefunctions are further bolstered by the recent observation that improved trial wavefunctions can also ameliorate the sign problem in free projection QMC calculations, where the constraints used to control the sign problem are either removed after some initial propagation time, or not employed at all.*"^{21,26}

7. May you break down the sources of your systematic errors QC-AFQMC errors compared to the exact results? How much is this from an inaccurate trial wave function? How much is this from potential measurement inaccuracies (if significant)? Are there other sources?

We thank the referee for their interest in this point. We know that the error from measurement inaccuracies is small because if we perform the AFQMC calculations with the "ideal" trial wavefunction (i.e., the trial wavefunction obtained without noise) we achieve similar accuracy as those from noisy experiments. The deviation of our experimental QC-AFQMC energies from those of ideal QC-AFQMC energies is less than 1 kcal/mol except for one geometry of diamond (lattice constant of 3.240 Å), where we see still a small deviation (1.3 kcal/mol). The remaining error in the ideal QC-AFQMC energies compared to the exact results is due to the limitation of our simple quantum trial wavefunctions.

Prompted by the referee's comment, we made a number of changes to clarify this point. We

provide those AFQMC energies calculated using the noise-free trials in the Supplementary Information. We have also added them to the caption of Figure 2 and to the plots in Figure 3 so that the interested reader can notice them easily. We have added the following text explaining our conclusions with regard to the sources of error to paragraph spanning the right and left-hand columns of page 5: *“At the same time, QC-AFQMC performed using the idealized quantum trial produces results comparable to our experimental energies, suggesting that the error in our QC-AFQMC energies is mainly due to the use of an insufficiently accurate trial wavefunction, rather than experimental error.”*

8. What is the actual time and resource expenditure for performing these quantum measurements?

We thank the reviewer for encouraging us to share this data with the reader. For each randomly chosen Clifford unitary, the repeated state preparation and measurement for shadow tomography takes about a second. For H_4 and N_2 , we took 15,000 Clifford measurements, which translates to about four hours of circuit execution time. For diamond, we performed 50,000 Clifford measurements, taking roughly 14 hours. These timings will necessarily become better as our quantum processors improve. We have added this information to the first paragraph of Supplementary Information E: *“It took approximately one second of device time to load each circuit and perform the repeated execution and measurement. The loading time is much larger than the time required to perform the measurement and reset, which is itself much larger than the time required to execute the circuit. We expect these timings to improve with future versions of our quantum processors. Furthermore, a larger chip would support parallel execution of multiple experiments. As a result, it took about four hours for H_4 and N_2 and 14 hours for diamond.”*

Minor Edits

1. Equation 1’s equations would be clearer if they were combined.

We agree with the referee and we had initially combined the two equations. However, we later changed it to its current form because we found it useful to reference $|\Psi(\tau)\rangle$ in Figure 1 and wanted to define this expression for the reader in Eq(1).

2. In the abstract, this sentence could be made less winding: ”...popular variational quantum eigensolver in terms of potential towards the first practical quantum advantage in ground state many-electron calculations.”

We thank the referee for this point. We changed the corresponding sentence by replacing it with the following: *“Compared with the variational quantum eigensolver, our new hybrid quantum-classical computational paradigm offers an alternative path towards achieving a practical quantum advantage for the electronic structure problem without demanding exceedingly accurate preparation and measurement of the exact ground state wavefunction.”*

3. This phrase is misworded: "...quantity that is directly is the ratio..."

We thank the referee for this point. The pertinent sentence (the last full sentence of the last full paragraph on the left-hand side of page 3) now reads "*Furthermore, our algorithm naturally achieves some degree of noise robustness explained in SI D6 because the quantity directly used in QC-QMC is the ratio between overlap values, which is inherently resilient to the estimates of the overlaps being rescaled.*"

4. Under Table 1, I believe you want to define QT as quantum trial in the caption, but didn't explicitly do so and vacillate between using quantum trial and QT in your Table and descriptions. Moreover, you could do a better job explicitly stating that the quantum trial is the wave function generated on your quantum computer. This was not clear to me for over a paragraph.

We thank the referee for this point. We changed QT to Q. trial, which is defined in the Table 1 caption.

5. Given the discussion of coupled cluster expansions, you should cite Sandeep Sharma's recent papers on free projection AFQMC with a coupled cluster trial, and its improved accuracy.

We thank the referee for this point. We added the citation to the sentence mentioning unbiased approaches. In the course of addressing one of the referee's above points (substantive question #6), we also added a sentence to the second-to-last paragraph of Supplementary Information F that mentions this work.

Referee #2

A new quantum monte carlo (QMC) approach for quantum computing in quantum chemistry has been developed, this methodology has not been used in near term quantum computing before, and what is even more impressive, has not been implemented on the actual hardware before. The methodology has a relevant place for benefiting from a quantum computer by introducing more accurate trial wavefunction ansatzes (e.g. based on unitary coupled cluster techniques) and evaluating classically challenging overlaps between walkers' wavefunctions and the trial wavefunction. This computational approach can be compared with previously developed the variational quantum eigensolver (VQE) where similar trial wavefunction ansatzes can be used. The advantage of the current approach is that it does not require a challenging optimization of trial wavefunction parameters and yet seems to provide more accurate results (although I have a question regarding this below). The considered chemical systems are indeed challenging for state-of-the-art classical methods and the obtained results are indeed impressive and will be of interest for the scientific community. Overall, the manuscript is written quiet clearly and can be understood by a general reader, but there are few places where more precise statements are needed (more detailed below).

We thank the referee for the accurate overview of our work and for sharing with us their positive opinions on our work.

I think the paper has a potential to be published in Nature after addressing a few questions

below:

1) Trial wavefunction within result on Fig. 3a, was it obtained with or without optimization of parameters (amplitudes for generators) in the VQE fashion? I suspect it is without optimization, then its results are not a good measure for advantage of the QMC approach over VQE. It would be great to see the comparison between QMC and VQE using the same ansatz for the trial function for some system even using a classical simulator. What is not clear for me is whether QMC surpasses VQE in accuracy for the same ansatz, but to answer this, one needs to optimize the ansatz parameters in the VQE case.

We thank the referee for this excellent question. This is an important point to make which we overlooked in our original main text. The trial wavefunctions used in our experiments involve parameters that were pre-optimized classically (although in principle we could have run the full VQE loop to optimize those ansatzes on the quantum computer). The trial wavefunction used in Fig. 3a (i.e., N_2) is a wavefunction called perfect pairing that can be optimized efficiently classically. Therefore, given the same quantum resources (i.e., the number of qubits), the best possible VQE results are the variational energies of our trial wavefunction in the limit of noise-free circuit executions. The raw data of such noiseless variational energies are now provided in Table S10 in the Supplementary Information. Idealized VQE results can be better than our reported noisy VQE results by at most 4.5 kcal/mol at one geometry, which is a minor improvement on the plotted energy scale in Fig. 3a (and our QC-QMC results are much better than this). To explicitly mention this, we added the following sentence to the end of the last full paragraph on the left-hand side of page 5: *“Idealized (i.e., noiseless) VQE experiments for the same trial wavefunction would yield similar results to our quantum trial results Fig. 3 (a) (within 4.5 kcal/mol), which are much worse than our QC-AFQMC results.”*

2) As was pointed out by the authors they didn't reach the chemical accuracy for N_2 in all geometry configurations, which is a challenging system, especially considering that the one-electron basis was cc-PVTZ, but what I am wondering is what would be needed to overcome this problem at least in theory? What I feel is missing in the manuscript (& supplementary info) is a discussion on what are the conditions for obtaining the exact full-CI energy via QMC approach?

We thank the referee for this question. In the limit where our trial wavefunction becomes exact QC-AFQMC recovers the exact full-CI energy. We mention this limit both in the main text and Supplementary Information B 2. We have tweaked the sentence in the main text which points this out to be more direct (on the right-hand side of page 2): *“Imposing these constraints necessarily introduce a potentially significant bias in the final ground state energy estimate which can be removed in the limit that the trial wavefunction approaches the exact ground state.”* We have also added some text to this paragraph to clarify the subtlety that we also typically expect the QMC energy to be more accurate than the energy of the bare trial wavefunction (this text is quoted in our response to the point directly below).

We now additionally mention an alternative strategy that one can pursue to recover the exact full-CI energy, namely, releasing the constraints on the calculation. Releasing the constraints removes the bias at the expense of suffering an uncontrolled sign problem. Depending on the

quality of the trial wavefunction and the computational effort that one is willing to expend, it is possible to recover the exact ground state energy using this approach. We have added the text “*Alternatively, the bias can be removed by releasing the constraints during propagation at the expense of suffering an uncontrolled sign problem.*”²⁶ to the end of the paragraph on the right-hand side of page 2 to mention this, and we also raise this point again in the second-to-last paragraph of Supplementary Information F.

Some qualitative statements are made that walkers wavefunctions and the trial wavefunction affect results, but it is not clear to me what limits the accuracy in the current approach? Technically, if we have any trial wavefunction that is not orthogonal to the exact wavefunction we should get the right answer, but in reality there are limitations on the walkers to be single determinant, which limits the overall accuracy, I assume. Some clear discussions of these questions are needed, they should address the question what one should do to systematically converge to the exact answer. Like in VQE one can make exponentially large ansatz to converge to the exact answer.

We thank the referee for urging us to make this discussion clearer to the reader. Like in VQE, QC-QMC results necessarily improve and ultimately reach the exact ground state energy as we improve the trial wavefunction. However, while the approximation in QC-QMC is controlled by the quality of the trial wavefunction, the QC-QMC result is significantly more accurate (converges faster) than the approximation in the trial wavefunction. Thus, the magnitude of the bias introduced by the constraints depends solely on the trial wavefunction and does not depend on the form of the walker wavefunction. We have modified the text on the left-hand side of page two to communicate this and to address related comments by the first referee. The relevant sentences now read: “*These constraints (which include prominent examples such as the fixed node^{17,23} and phaseless approximations^{24,25}) are imposed by demanding that the overlaps of the trial wavefunction ($|\Psi_T\rangle$) with the stochastic samples ($|\varphi_i(\tau)\rangle$) remain positive during the imaginary-time propagation. While constrained QMC calculations are typically much more accurate than those using the bare trial wavefunction directly (see Fig. 1(b) bottom), the remaining bias of the constrained QMC results is wholly determined by the choice of trial wavefunction.*”

Furthermore, similar to VQE, searching for compact and systematically improvable trial wavefunctions in QC-QMC is still a research-level question that we hope to explore shortly. We mention examples of trial wavefunctions to increase the accuracy in our main text, such as unitary coupled-cluster states, qubit coupled-cluster wavefunctions, and the multiscale entanglement renormalization ansatz. We also added a mention of wavefunctions constructed by adiabatic state preparation. The unitary coupled-cluster state and its simplified versions are good candidates for systematically improving the trial wavefunction by adding more excitation operators. In the context of QC-QMC, we have the advantage of being able to leverage the improved wavefunctions more effectively than VQE. One can see evidence of this property in the (b) and (c) subpanels of Figure 2, where the QC-AFQMC energies converge more quickly and to more accurate values than the energies of the bare trial wavefunctions. We have added the sentence “*Notably, we achieve this accuracy even in the larger basis, where the variational energy of the quantum trial in the absence of noise is far from exact.*” to the paragraph that spans

the left-hand and right-hand columns of page 4 to draw the reader's attention to the fact that an inexact trial wavefunction can be sufficient to obtain good accuracy.

3) Weights update: this part is not clear, how the weights are selected and why needs to be explained more (at least in an appendix). For example, around eq B5, I would suggest to state clearly what is the equation for the weight at $\tau + d\tau$ as a function of previous time weights, and explain why such an update is used.

We thank the referee for this question. This was a mistake on our side in that Eq.S5 (previously Eq.B5) should define the weight update rules more clearly. Eq.S5 now reads

$$w_i(\tau + \Delta\tau) = w_i(\tau) \times |S_i(\tau)| \times \max(0, \cos \theta_i(\tau))$$

4) What is the role of the Hubbard-Stratonovich transformation in imaginary time propagation? Why is it needed on top of the trotterization?

It is to turn the many-body imaginary time propagator (i.e., e^{-tH}) into a single-particle imaginary time propagator coupled to fluctuating Gaussian random variables called the auxiliary fields. This is convenient because the action of a single-particle operator on a Slater determinant yields another Slater determinant. As this is a key technical detail for AFQMC (not necessarily for QC-QMC in general), we have added the following sentence in the AFQMC-specific section of the Supplementary Information (Supplementary Information B2): *"In other words, the Hubbard-Stratonovich transformation turns the many-body imaginary time propagator into single-particle imaginary time propagators coupled to Gaussian random variables called auxiliary fields."*

5) In selection of the trial wavefunction ansatz, the authors do not mention any non-UCC-derived approaches, even though it was shown that qubit-based techniques can be more efficient (fewer 2-qubit gates and lower circuit depth) than those obtained from the fermionic algebras. I think this needs to be mentioned.

We thank the referee for this suggestion. Indeed, qubit-based approaches such as qubit coupled-cluster theory should be mentioned. We now cite the pertinent work in the following sentence in the main text (which occurs in the paragraph spanning pages two and three): *"There is currently no efficient classical algorithm to estimate (to additive error) the overlap between $\varphi_i(\tau)$ and certain complex quantum trial wavefunctions Ψ_τ such as unitary coupled-cluster with singles and doubles,²⁸ qubit coupled-cluster methods,²⁹ wavefunctions constructed by adiabatic state preparation,³⁰ or the multiscale entanglement renormalization ansatz,³¹ even when $|\varphi_i(\tau)\rangle$ is simply a computational basis state or a Slater determinant."*

6) Choosing the shadow tomography over introducing the transformations into the eigen-frame of P_\pm operators has not been justified. It would be interesting to know why the authors believe that random circuits are more efficient than those that bring P_\pm operators into the Z-basis (computational basis).

Diagonalizing the P_{\pm} operators is essentially equivalent to performing the ancilla-free version of the Hadamard test for the wavefunction overlap that we alluded to at the beginning of Supplementary Information C2. We have added two sentences to the second paragraph of Supplementary Information D2 to draw the reader's attention to this connection: "*While we use these observables to construct our measurement protocol using shadow tomography, we note in passing that they are also related to the alternative approach based on the ancilla-free Hadamard test.⁶⁰⁻⁶² By preparing $|\tau\rangle$ and diagonalizing P_{\pm} , one can measure $\langle\phi_i|\Psi_T\rangle$ without using an ancilla to control the state preparation circuits for $|\phi_i\rangle$ or $|\Psi_T\rangle$.*"

Using the Hadamard test would offer some advantages compared to our randomized measurement approach, removing the need for the expensive post-processing in QC-AFQMC. However, the circuits that diagonalize each of the P_{\pm} operators depend on the walker parameters. This would force us to interactively query the quantum processor for each walker at each timestep. This overhead, especially when we consider the impact of the communication latency and the difficulty in tightly coupling the quantum and classical computation, pushed us to consider an alternative based on shadow tomography. The shadow tomography protocol allows us to characterize the quantum trial wavefunction before determining the walker parameters using a collection of randomized measurements. We can then approximate the results of an interactive protocol by post-processing the resulting classical shadow, simplifying the experiment significantly.

Referee #3 Review

In "Unbiasing Fermionic Quantum Monte Carlo with a Quantum Computer", by Huggins et al. they present a new way of utilize NISQ era devices. The hybrid- quantum classical approach leverages the intellectual technology of Monte Carlo and the hardware technology of Google Quantum AI Labs. From the Monte Carlo vantage, the paper relies on auxiliary field quantum Monte Carlo (AFQMC) and aim to allieviate the errors made by the phase less approximation. The walkers are updated following a phaseless approximation whereby negative weight walkers can be removed. The error due to the phaseless approximation depends on the trial wave function. This leads to the quantum computational part of the algorithm. The quantum algorithm for this methods requires measuring overlaps between the walkers and the quantum computer's trial wave function. This is a novel concept that I was excited to learn about and is precisely the type of manuscript that should be published in Nature.

We are glad to hear that the reviewer found our work novel and worthy of being published in Nature.

The measurement AFQMC scheme does not require any feedback loops between the quantum device and the classical hardware. This scheme still relies on preparing a good quantum state (which is QMA-complete). They utilize a specific classical ansatz (perfect pairing) and, in some cases, perform further variational optimization of the state. The obstacle of state preparation still looms over this approach. Despite the advances provide, this is not a silver bullet for demonstrating quantum supremacy via simulation methods. (The authors thoughtfully address this point the conclusions and in Appendix F.)

Indeed, solving the ground state exactly is generically *hard*. Our goal was to utilize the best available quantum states to unbiased accurate state-of-the-art constrained QMC methods. As the referee points out, we have tried to carefully document the many possible considerations that affect where our QC-QMC algorithm may achieve a practical quantum advantage over other state-of-the-art classical methods.

Also of interest is the use of the relatively new technique of shadow tomography. This allows the authors to reach state-of-the-art quantum simulations of molecular systems larger than all previous quantum-classical hybrid schemes.

While the shadow tomography technique is not strictly necessary for the workflow of QC-QMC (since one can instead use a Hadamard test), this technique certainly makes QC-QMC easier to implement on near-term devices. We are happy to hear that the referee found our use of shadow tomography in this work interesting.

I found the paper well written and well thought through. The paper is a good representation of the modern efforts in quantum computing and this paper fits into the Nature magazine goals quite nicely. I saw no typos and the paper, its logic, and results all seems correct.

We thank the referee for this comment.

One question I have: other Monte Carlo methods (VMC, DMC, and FCIQMC) aren't commented on extensively. I am not as familiar with AFQMC as the other methods. The other approaches don't require extracting overlaps. It isn't clear from reading the main text and perusing the many appendices if there is direct application to these other methods QMC methods. For instance, the trial wave function determines the nodes of DMC but its not clear that extracting the overlaps will help improve the nodal structure.

This is a good question and we thank the referee for raising it. This paper focuses on the constrained projector QMC methods, so we did not talk about the VMC method (since it is not a projector QMC approach) in the main text. Similar strategies explored in this work may be helpful in VMC, but we have not thought about this question deeply at present. We mention this possibility in Supplementary Information A: "*In passing, we note that variational QMC may benefit from some of the techniques proposed in this work, which could be interesting to study further in the future.*"

At the time of writing this manuscript, we did not discuss FCIQMC because while it is a projector QMC method, it usually does not employ any constraints to cope with the fermionic sign problem. Near the completion of work, a fixed-node variant of FCIQMC appeared, which we now mention and cite in Supplementary Information A and Supplementary Information C4.

Furthermore, the QC-QMC algorithm presented in our main text should apply to fixed-node DMC. While it is not the conventional viewpoint, the fixed-node DMC approach evaluates the overlap between $\langle \Psi_T |$ and a walker wavefunction $|\mathbf{r}\rangle$ (a vector in \mathbb{R}^{3N} where N is the number of electrons). This task precisely corresponds to evaluating Ψ_T in real space, which is how practical DMC programs operate.

Since this is an important technical point to include in our supplementary materials, we added a subsection with the following sentences: “As mentioned in SI B, our QC-QMC algorithm can be specialized to other projector QMC methods. Here, we provide brief details on other QC-QMC methods, lattice GFMC, FCIQMC, and DMC. In lattice GFMC, walkers are represented by a determinant, $|n\rangle$, in the computational basis. Therefore, the fixed-node constraint utilizes the overlap value, $\langle\Psi_T|n\rangle$, which can be estimated by using the quantum computer. Similar strategies can be explored in fixed-node FCIQMC.⁵⁰ In DMC, walkers are represented by a vector in \mathbb{R}^{3N} , $|\mathbf{r}\rangle$. The fixed-node constraint in DMC then uses the overlap value, $\langle\Psi_T|\mathbf{r}\rangle = \Psi_T(\mathbf{r})$, which can be obtained from the quantum computer.”

One mild suggestion I have: in the main text, the description of the technique is quite terse. It is unclear from reading the main text how the hybrid algorithm work (even at a cursory level). Equations B3 and B6 seem to underlie the main idea of the whole paper. I think including them e.g. in the caption for Figure 2 or before turning to ”Results and discussion” would be ideal from a reader’s perspective.

We thank the referee for this point. We realize that in our desire to present an accessible high-level overview we have neglected some helpful specifics that might benefit the interested reader. We now include some text in the final paragraph before the “Results and discussion” section that explains some of these specifics and refers the reader directly to the key equations that the referee highlighted: “In AFQMC, the $|\varphi_i(\tau)\rangle$ take the form of Slater determinants in arbitrary single-particle bases, allowing us to express the energy estimator (presented in Eq. (S3)) in terms of a modest number of wavefunction overlaps we can evaluate efficiently on the quantum processor (see SI C2). The phaseless constraint is imposed to control the sign problem and likewise only requires calculating the overlaps between $|\Psi_T\rangle$ and $|\varphi_i(\tau)\rangle$, as detailed in Eq. (S6).“

Overall recommendation: publish with optional edits.

Referee #4

The manuscript by Huggins et al. provides a new take on the use of quantum computer prototypes as co-processors in quantum chemistry calculations. The authors show that the use of a quantum device as part of a quantum Monte Carlo calculation performed on a classical computer can, when combined with a few additional (classical) tricks, provide an advantage over the common classical approach. The results are contrasted to a number of other common approaches in quantum chemistry, both classical (CCSD(T)) and quantum (VQE) and surpass these.

We thank the referee for a good overview of our work.

The paper is written rather technical and comprehensive, visible not only in the large number of references, but also the very extensive appendix, which is frequently referenced. This style of presentation, while technically very complete and accurate, makes it harder to follow the presentation and somewhat obscures the achievement of this new algorithmic method.

We thank the referee for their comments. We would like to point out that our extensive list of references is mainly for our Supplementary Information (originally Appendices). The number of references in the original manuscript was 45 (smaller than the maximum number, 50, of references mentioned in the journal guideline), and the revised main text now has 48 references. We also separated out the Supplementary Information from the main text. Nature has a broad readership, so we took special care in our presentation to reduce unnecessary technical details in the main text while providing necessary information for experts in the Supplementary Materials. We note that Referees #1, #2, and #3, had generally positive comments regarding the readability and clarity of both the main text and the Supplementary Information. However, we would be happy to try to improve our exposition in response to more specific feedback.

The frequent reference to the present work being the largest quantum chemistry simulation ever performed on quantum hardware does not signal a major breakthrough, however, but rather an incremental improvement.

We believe that our work represents a significant advancement in this field for several reasons. The prior largest quantum computation of chemical systems, including electron correlation, was performed in 2017 and was limited to 6 qubits.¹³ Their work uses a quantum-classical hybrid algorithm called the variational quantum eigensolver (VQE) approach that has been the main workhorse of NISQ applications in chemistry. However, over the past four years, no research groups have reported VQE calculations using more than six qubits that tackle correlated electronic structure. This has not been for a lack of effort; there have been scores of papers reporting experimental VQE realizations for chemistry in that period. Rather, the practical limitations of VQE on today's devices have limited the scope of these experiments. In our work, we developed a novel hybrid algorithm completely different from VQE. We presented accurate quantum computations of correlated chemical systems up to 16 qubits, more than doubling the prior record. The size of the experiment is not the novelty per se, but rather indicative of the qualitative advance that our novel methods enable.

Moreover, due to the virtual correlation technique developed in our work, we can obtain accurate electron correlation energies in Hilbert spaces far larger than the number of qubits we've used would suggest. This technique is a remarkable development because all prior VQE-driven methodologies were limited to a small number of orbitals (fewer than 10), making them far from competitive with state-of-the-art classical methods which can handle hundreds of orbitals. Our QC-QMC approach can handle hundreds of orbitals while limiting the quantum resources to a much smaller active space without any additional measurement overheads. Therefore, we can compare QC-QMC results with state-of-the-art classical methods for challenging systems with many electrons and orbitals far beyond the reach of any other proposed NISQ algorithms. This point has been demonstrated by our experimental treatment of three representative chemical systems.

Due to the reasons stated above, and the noise resilience properties that the referee discusses below, we believe that our work represents a major algorithmic advancement and experimental achievement and can not be viewed as an incremental improvement. Indeed, as the reviewer notes, it is an entirely novel method. However, if the reviewer is commenting more on advances in the hardware itself - we would agree. Prior experiments have been published in Nature and other journals using the same generation of Sycamore hardware platform. While the hardware is

absolutely state-of-the-art, the major advances here compared to prior work are entirely due to the new algorithms, rather than due to intrinsically better hardware.

In fact, as shown and openly stated in the paper, the bare results and noise levels from the machine prevent it from producing reliable results - it is simply the structure of the (mostly classical) algorithm calling the quantum device, that makes the results useable in the examples demonstrated.

The point raised by the referee actually emphasizes how noise-resilient our QC-QMC algorithm is. Achieving substantial noise resilience has been a primary goal of NISQ algorithm development and is necessary for adequate performance on any application. In QC-QMC, we deliberately delegate only the classically difficult tasks (i.e., the measurement of overlap) to the quantum computer. The rest of the QMC calculations are done on the classical computer. It is designed this way so that quantum and classical computers are assigned tasks that each can perform well, and some practical quantum advantage may still be obtained.

The power of QC-QMC is that the final QC-QMC result is relatively insensitive to the often mediocre bare results (i.e., the quantum trial under circuit noise). However, the approach still requires a good degree of quantitative accuracy from the overlap measurements in order to perform well. As shown in Fig. 2 (b) and (c), when the error is sufficiently large the additional robustness offered by our approach is not enough to produce good results. If the quantum computer performs poorly at its job, e.g., by under-sampling the random measurements, the overall QC-AFQMC does not reach chemical accuracy. Only after the quantum computer performs enough random measurements does QC-AFQMC perform well. We have also demonstrated that the use of the quantum trial wavefunctions yields an improvement over a simple and typical choice of the classical trial wavefunction, even in the presence of noise. This strengthens our case that the quantum computer is providing useful data to the classical algorithm.

As Figure 3.b shows, larger systems (despite only including 2 atoms and 26 orbitals - much less than the 60 orbitals in 3.a) no longer shown chemical accuracy.

We would like to emphasize the following two points.

Our QC-QMC is systematically improvable by improving the quality of the quantum trial wavefunction. For proof-of-concept, we chose the simplest possible trial wavefunction and implemented it on the quantum processor. If we were to use more sophisticated trial wavefunctions, the remaining bias would be further reduced. Exploring more sophisticated quantum trials as our NISQ devices improve is one of our immediate research directions.

Furthermore, other state-of-the-art classical approaches perform similarly or worse than QC-QMC. We would like to point out that carrying out other NISQ algorithms such as VQE on 26 orbitals (which would correspond to 52 qubits) is currently impossible. The fact that QC-QMC produces results that are competitive to state-of-the-art classical approaches is genuinely remarkable and represents a breakthrough in this area.

Due to these reasons, we do not think that the point raised by the referee is necessarily a concern. Instead, it should be viewed as providing exciting opportunities for future exploration laid out by our work.

In summary, while the specific algorithmic workflow using the quantum chip is novel, there

is nothing surprising or a breakthrough in these results.

We respectfully disagree with the referee. We have provided multiple reasons above that support our assertion that our work represents a considerable advancement in this field.

The heavy lifting remains on the classical computing side and the quantum chip's noisy data is evaluated in a way that makes the calculation less susceptible to noise.

We agree with the referee on this point and we would like to emphasize again that this is precisely the main objective of quantum-classical hybrid algorithms (or NISQ algorithms)! The key design principle is that we want the quantum computer to perform tasks difficult for the classical computer and vice versa. While doing so, our QC-QMC algorithm is much less susceptible to noise than other hybrid algorithms such as VQE, which is the most important reason that we believe that our work will initiate a paradigm shift in this field. To emphasize this point, we added the following sentence in the abstract: *“Compared with the variational quantum eigensolver, our new hybrid quantum-classical computational paradigm offers an alternative path towards achieving a practical quantum advantage for the electronic structure problem without demanding exceedingly accurate preparation and measurement of the exact ground state wavefunction.”*

Lastly, given the small number of gates (Fig 2.a) and qubits (8-16) used in these experiments, I do wonder, why the authors do not simulate their quantum computer to make a statement about suitability and scalability under reduced noise levels and improved hardware.

We thank the referee for this question. We agree with the reviewer that it is a small enough circuit to simulate classically. In fact, we have noise-free results that assume perfect circuit execution (i.e., no circuit noise) and an infinite number of samples in shadow tomography (i.e., no sampling noise). We did not study how the noise influences our reported QC-QMC results in detail because what we obtained directly from the Sycamore quantum processor is already remarkably close to the noise-free results. In order to highlight this similarity for other readers interested in the extent to which reduced levels of noise would improve performance we now include these noise-free results in the caption of Figure 2 and in the plots of Figure 3. We have also added the following text to the paragraph that spans the two columns of page 5: *“At the same time, QC-AFQMC performed using the idealized quantum trial produces results comparable to our experimental energies, suggesting that the error in our QC-AFQMC energies is mainly due to the use of an insufficiently accurate trial wavefunction, rather than experimental error.”*

To address the reviewer's question further, we added more comprehensive noise-free results in Supplementary Information E for all three chemical systems. Since these are all small energy scales (less than 1.34 kcal/mol in all cases), we did not find extensive numerical studies of the impact of different types of noise necessary for the goal of this paper. We had initially performed some preliminary studies with a coarse-grained noise model based on single-qubit depolarizing noise. Ultimately, we did not find these numerical studies informative enough to be included in

the final paper due to the substantial difference between the simplified noise model and the real noise processes of the device. In the future, it would be interesting to try to better understand the noise resilience of our approach analytically and by performing simulations under various noise models.

Ultimately, I cannot recommend publication in Nature and would recommend to submit to a more specialized quantum journal such as npj quantum information.

We thank the referee for the input provided, but we respectfully disagree with their overall assessment. We believe our work presents the development of a novel hybrid algorithm (QC-QMC) and successful implementation and demonstration of the algorithm on the Sycamore quantum processor. QC-QMC is fundamentally different from the popular VQE algorithm, and it enables the most scalable quantum computation of correlated chemical systems to date. It already achieves competitive accuracy compared to state-of-the-art classical electronic structure methods for challenging chemical systems containing up to 120 orbitals, which has not been possible by other NISQ algorithms. Our work thus represents a considerable innovation in this field that will bring a paradigm shift in many research areas, making it well-suited for publication in Nature.

Reviewer Reports on the First Revision:

Referee #1 (Remarks to the Author):

After reading all of the responses to my questions and others, I am satisfied with the revisions the authors have made. I particularly appreciated their well-thought responses to my question regarding the exponential scaling that may accompany measuring the overlaps. While the paper can't be neatly summarized into the one-sentence explanation of novelty and/or impact for laymen that many Nature papers can, the paper does present a novel idea and is accompanied by very thorough Appendices that provide a wealth of information to the larger community. I therefore support its publication at this point.

Referee #2 (Remarks to the Author):

All my questions were addressed sufficiently for publication of this manuscript.

Referee #3 (Remarks to the Author):

I have read the authors responses and their revised draft. Their responses were thoughtful and I believe it should be published as is.

Referee #4 (Remarks to the Author):

I thank the authors for their thorough and thoughtful reply as well as the changes made in response to my comments. While I do stand by my original review, I can see the potential value for Nature's readership in the other referees' comments (who appear closer to the application / user side) as well as the elaborate responses of the authors. As such, I would like to provide only a few comments on the first page to further sharpen the message and get the achievement of the paper across without giving in to the numbers game ("largest chemistry simulation in terms of number of qubits").

To me the main achievement is not (yet) a quantum advantage, but a novel, more optimal exploitation of the quantum resource in a classical computational workflow, which achieves competitiveness with state-of-the-art classical methods for small problems even with the current noisy devices.

Here are my suggestions:

"Abstract"

- * Greatest challenges in computational science —> greatest computational challenges
- * Largest chemistry simulations performed [on] quantum computers —> with the help of
- * I suggest removing the bracket (more than doubling...) and writing something akin to "for the first time obtaining accuracy with ... on a quantum device without error mitigation." This is to signal how this new approach can better deal with noise in pre-error correction systems and implicitly point to luck and post-selection in the other papers, which appear not to have been necessary here?
- * Add "popular" or "mainly pursued" in front of VQE to better contrast that this is not just another take on the variational approach

Paragraph 1

- * Polynomially with system size but which are [only] approximate
- * obtained by these methods [often is] unsatisfactory

End of paragraph 2: Using this novel approach, our experimental demonstration surpasses...

Author Rebuttals to First Revision:

Referee #1:

After reading all of the responses to my questions and others, I am satisfied with the revisions the authors have made. I particularly appreciated their well-thought responses to my question regarding the exponential scaling that may accompany measuring the overlaps. While the paper can't be neatly summarized into the one-sentence explanation of novelty and/or impact for laymen that many Nature papers can, the paper does present a novel idea and is accompanied by very thorough Appendices that provide a wealth of information to the larger community. I therefore support its publication at this point.

We thank the referee for supporting the publication of our article.

Referee #2:

All my questions were addressed sufficiently for publication of this manuscript.

We thank the referee for supporting the publication of our article.

Referee #3:

I have read the authors responses and their revised draft. Their responses were thoughtful and I believe it should be published as is.

We thank the referee for supporting the publication of our article.

Referee #4:

I thank the authors for their thorough and thoughtful reply as well as the changes made in response to my comments. While I do stand by my original review, I can see the potential value for Nature's readership in the other referees' comments (who appear closer to the application / user side) as well as the elaborate responses of the authors. As such, I would like to provide only

a few comments on the first page to further sharpen the message and get the achievement of the paper across without giving in to the numbers game (“largest chemistry simulation in terms of number of qubits”).

To me the main achievement is not (yet) a quantum advantage, but a novel, more optimal exploitation of the quantum resource in a classical computational workflow, which achieves competitiveness with state-of-the-art classical methods for small problems even with the current noisy devices.

We thank the referee for the comments.

Here are my suggestions:

“Abstract”

* Greatest challenges in computational science → greatest computational challenges

We thank the referee for this point. We have incorporated this suggestion into our manuscript.

* Largest chemistry simulations performed [on] quantum computers → with the help of

We thank the referee for this point. We have incorporated this suggestion into our manuscript.

* I suggest removing the bracket (more than doubling...) and writing something akin to “for the first time obtaining accuracy with ... on a quantum device without error mitigation.” This is to signal how this new approach can better deal with noise in pre-error correction systems and implicitly point to luck and post-selection in the other papers, which appear not to have been necessary here?

We thank the referee for this point. We have incorporated this suggestion into our manuscript.

* Add “popular” or “mainly pursued” in front of VQE to better contrast that this is not just another take on the variational approach

We thank the referee for this point. We have incorporated this suggestion into our manuscript.

Paragraph 1

* Polynomially with system size but which are [only] approximate

We thank the referee for this point. We have incorporated this suggestion into our manuscript.

* obtained by these methods [often is] unsatisfactory

We thank the referee for this point. We have incorporated this suggestion into our manuscript.

End of paragraph 2: Using this novel approach, our experimental demonstration surpasses...

We thank the referee for this point. We have incorporated this suggestion into our manuscript.